# Preferential lactate metabolism in the human brain during exogenous and endogenous hyperlactataemia

Jodie L. Koep[1,2] , Jennifer S. Duffy[3], Jay M. J. R. Carr[4], Madden L. Brewster[1,2], Jordan D. Bird[1,2,5], Justin A. Monteleone[1,2], Tenasia D. R. Monaghan[1,2], Hashim Islam[1,6], Andrew R. Steele[1,2], Connor A. Howe[1,2], David B. MacLeod[7], Philip N. Ainslie[1,2] and Travis D. Gibbons[8]

[1] School of Health & Exercise Sciences, University of British Columbia – Okanagan, Kelowna, BC, Canada
[2] Centre for Heart, Lung and Vascular Health, University of British Columbia – Okanagan, Kelowna, BC, Canada
[3] School of Physical Education, Sport & Exercise Sciences, University of Otago, Dunedin, New Zealand
[4] Institute of Mountain Emergency Medicine, Eurac Research, Bolzano, Italy
[5] Division of Critical Care Medicine, Department of Medicine, Faculty of Medicine, Vancouver General Hospital, University of British Columbia, Vancouver, BC, Canada
[6] Centre for Chronic Disease Prevention & Management, Faculty of Medicine, UBC Okanagan, Kelowna, BC, Canada
[7] Department of Anesthesiology, Duke University Medical Center, Durham, NC, USA
[8] Department of Biological Sciences, Northern Arizona University, Flagstaff, AZ, USA

The peer review history is available in the Supporting Information section of this article (https://doi.org/10.1113/JP289216#support-information-section).

The Journal of Physiology

**Abstract figure legend** Trans-cerebral blood sampling of arterial and jugular venous blood samples, combined with duplex ultrasound, were used to calculate cerebral metabolic rates of glucose ($CMR_{Glc}$) lactate ($CMR_{iLac}$) and oxygen ($CMR_{O_2}$). Thirteen healthy adults completed protocols involving: (1) intravenous sodium lactate infusion (exogenous hyperlactataemia) and (2) cycling exercise (endogenous hyperlactataemia), to achieve matched arterial lactate concentrations ($\sim$0, $\sim$4 and $\sim$8 mmol/l, respectively). Pie charts represent the relative contribution of glucose ($CMR_{Glc}$) and lactate ($CMR_{iLac}$) to total cerebral oxidative metabolism ($CMR_{O_2}$) at rest (0 mmol/l) and during hyperlactataemia at 4 and 8 mmol/l, respectively. Despite physiological differences between protocols, cerebral lactate oxidation increased and oxidative glucose metabolism decreased in a dose-dependent manner, with no change in $CMR_{O_2}$. At the 8 mmol/l concentration, lactate accounted for $\sim$24% of total cerebral oxidative metabolism, indicating a substrate switch toward lactate, and a sparring of available glucose. These findings were irrespective of whether lactate is delivered exogenously or produced endogenously.

**Abstract** At rest, glucose serves as the brain's primary oxidative substrate; however, when circulating lactate is elevated, lactate oxidation increases. Whether this glucose-sparing effect differs when lactate is elevated via passive infusion *versus* exercise remains unknown. To address this, 13 healthy adults (six females) completed protocols of: (1) intravenous sodium lactate infusion (exogenous hyperlactataemia); and (2) cycling exercise (endogenous hyperlactataemia) to matched elevations in arterial lactate concentration ($\sim$4 and $\sim$8 mmol/l). Radial arterial and internal jugular venous sampling and measures of cerebral blood flow (CBF) were used to calculate cerebral metabolic rates of glucose ($CMR_{Glc}$), lactate ($CMR_{iLac}$), and oxygen ($CMR_{O_2}$). The exogenous infusion protocol resulted in a higher CBF compared to exercise ($P < 0.001$), despite causing systemic alkalosis ($P < 0.001$). During both protocols $CMR_{O_2}$ remained unchanged across increases in lactate concentrations ($P = 0.610$), while $CMR_{Glc}$ decreased (lactate, $P = 0.009$; condition, $P = 0.373$) and $CMR_{iLac}$ increased in a dose-dependent manner (lactate, $P < 0.001$; condition, $P = 0.972$). At an arterial concentration of 8 mmol/l, circulating lactate accounted for 24% of total cerebral oxidative metabolism. Elevated circulating lactate leads to preferential lactate oxidation and reduced glucose utilization, irrespective of whether lactate is delivered exogenously or produced endogenously.

(Received 5 May 2025; accepted after revision 4 September 2025; first published online 30 September 2025)

**Corresponding author** J. L. Koep: University of British Columbia, Okanagan, School of Health and Exercise Sciences 1147 University Way, Kelowna, BC V1V 1V7, Canada.    Email: jodie.koep@ubc.ca

## Key points

- The human brain relies primarily on oxidative glucose metabolism; however, with age and in many pathologies cerebral glucose metabolism declines; therefore, there is interest in investigating alternative fuel sources that can meet the high energetic needs of the brain.
- The present study investigates whether increased lactate availability exerts a glucose-sparing effect in the healthy human brain, and whether this effect is consistent across physiologically distinct states of exogenous (sodium lactate infusion) and endogenous (exercise-induced) hyperlactataemia.
- We assessed cerebral uptake and metabolism of glucose and lactate following exercise and lactate infusion, using simultaneous arterial and jugular venous blood samples, and Duplex ultrasound.
- Despite stark systemic physiological differences between conditions, cerebral glucose metabolism declined in proportion to increased circulating lactate irrespective of whether it is delivered exogenously or produced endogenously.
- These data provide clear evidence that lactate is preferentially oxidized by the brain when made available, helping preserve glucose for non-energetic roles.

## Introduction

The brain is fuelled almost exclusively by oxidative glucose metabolism throughout the human lifespan. With age and in many pathologies, cerebral glucose metabolism declines and, therefore, there is interest in investigating alternative fuel sources that can meet the high energetic

needs of the brain (Chen & Zhong, 2013; Costantini et al., 2008; Cunnane et al., 2011; Cunnane et al., 2016; Goyal et al., 2017). In the last 20–30 years, the role of lactate as an alternative or complementary fuel for the brain has been well-established (Smith et al., 2003; Van Hall et al., 2009). The blood–brain barrier (BBB), neurons and astrocytes all contain monocarboxylate transporters (MCTs) that allow lactate to pass from the circulation into the brain, as well as between different cell types and cerebrospinal fluid. Indeed, rodent and human data estimate the rate of lactate transport across the BBB to be 25–50% of that of glucose (Cremer et al., 1979; Knudsen et al., 1991). Moreover, *ex vivo* data shows cultured neurons bathed in a medium containing both glucose and 5–10 mmol/l lactate generate more electrical energy from lactate compared to glucose (Larrabee, 1995; Tabernero et al., 1996). Lactate has been shown to support synaptic function in hippocampal slices in the absence of glucose (Schurr et al., 1988). Evidence also indicates that neurons rely primarily on the oxidative metabolism of lactate, which is shuttled from astrocytes undergoing glycolytic metabolism (Pellerin & Magistretti, 1994).

In humans, increasing circulating lactate to only 4 mmol/l via sodium lactate infusion decreases cerebral glucose metabolism ($CMR_{Glc}$) as indicated by fluoro-deoxyglucose positron emission tomography (FDG-PET) (Smith et al., 2003). Endogenously produced lactate (i.e. intense exercise) is extracted and oxidized in proportion to its arterial concentration (Siebenmann et al., 2021; Van Hall et al., 2009). In both young and older humans, exercise-induced elevations in lactate decrease glucose oxidation as measured via FDG-PET (Green et al., 2025; Kemppainen et al., 2005). This evidence indicates that when lactate is made available to the brain, whether exogenously or endogenously, it spares some of the energetic requirements normally provided by glucose. No study to date has directly compared passive lactate infusion *versus* exercise, targeted to match elevated arterial lactate concentrations. It is difficult to infuse lactate at a rate that reaches circulating lactate concentrations achieved easily during moderate exercise (e.g. 4 mmol/l), owing to acid–base disturbances associated with high a sodium load (Miller et al., 2005). Indeed, hyperlactataemia caused by passive infusion elicits metabolic alkalosis (Miller et al., 2005), whereas exercise-induced

hyperlactataemia elicits metabolic acidosis (Margaria et al., 1933). Although these acid–base changes likely have marked influence on cerebral blood flow (CBF) regulation (Caldwell, Carr, et al., 2021), how these factors modulate the effect of lactate on $CMR_{Glc}$ is unknown.

In this study we sought to determine $CMR_{Glc}$ in response to comparable elevations in circulating lactate caused by: (1) sodium lactate infusion and (2) intense exercise, in healthy humans. We hypothesized that $CMR_{Glc}$ would decrease in proportion to increased circulating lactate concentrations and this would be independent of exogenous or endogenous hyperlactataemia.

## Methods

### Ethical approval

This study was approved by the clinical research ethics board of the University of British Columbia (no. H23-02303). All experimental protocols and procedures conformed to the standards set by the Canadian government Tri-Council policy statement for integrity in research, as well as the *Declaration of Helsinki*, including registration in a database (approval no. 77764).

A detailed verbal and written explanation of the procedures and measurements was provided to participants prior to providing written, informed consent. Experiments were conducted with medical support and by suitably qualified personnel. Healthy normotensive volunteers who did not require daily medication (excluding contraceptive medications) were recruited for the study. Exclusion criteria included current or former smokers, a known history of cardiometabolic or respiratory diseases, and the use of medications. Thirteen aerobically fit [maximal oxygen uptake ($\dot{V}_{O_2max}$): 46.9 ± 6.2 ml/min/kg] participants (age 28.2 ± 3.5 years, six females) were recruited.

### Experimental overview

A preliminary visit was conducted to determine individuals' lactate threshold and maximal aerobic power 1 week prior to the primary experimental testing

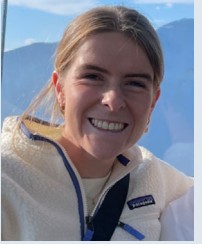

**Jodie L. Koep** is a Postdoctoral Fellow at the University of British Columbia, Okanagan. Her research and academic interests include cerebral vascular health, cardiometabolic physiology, and adaptive responses to exercise and environmental stressors. Her work focuses on understanding the mechanisms underlying vascular control during ageing and development. She is particularly interested in exploring the potential influence of exercise on vascular and metabolic function across healthy ageing and chronic disease.

session. Upon arrival, participants' body mass and height were measured. Following this, participants performed an incremental cycle ergometer test to exhaustion (Cosmed, Quark PFT, Rome, Italy), fitted with a face mask (Hans Rudolph, Shawnee, KS, USA) for assessment of lactate threshold and $\dot{V}_{O_2max}$.

For the experimental visit, participants arrived at the laboratory at 08.00 or 16.00 h, after refraining from alcohol for a minimum of 12 h. Participants maintained their normal caffeine intake and eating habits but did not consume any calories within 2 h of data collection. Participants were catheterized with radial artery and internal jugular bulb cannulas as detailed below, with ~2 h between catheterization and the start of the experimental protocol. Participants inserted a rectal thermistor to enable temperature correction of arterial blood gases using standard procedures (Severinghaus, 1966). Participants were also instrumented with a peripheral venous catheter, used as the site of lactate infusion during the passive hyperlactaemia protocol. This study followed a repeated measures design with all individuals completing the sodium lactate infusion protocol followed by 1 h of recovery and then the high-intensity interval exercise protocol to matched increases in arterial lactate concentrations between passive infusion and exercise of 4 mmol/l and 8 mmol/l stages, respectively.

## Catheterization

Ultrasound guided catheterization of a radial artery and internal jugular vein was performed using a sterile technique, under local anaesthesia (lidocaine, 2%). With participants resting supine, a 20G, 4.5 cm arterial catheter (Arrow, Markham, ON, Canada) was advanced into the radial artery, and a 16G, 16 cm central venous catheter (Cook Medical, Bloomington, IN, USA) was inserted into the internal jugular vein and advanced ~15 cm cephalad to the jugular bulb. Radial artery and jugular vein catheters were attached to in-line waste-less blood sampling systems for repeated trans-cerebral blood sampling (Edwards Lifesciences, TruWave VAMP, Irvine, CA, USA) (Ainslie et al., 2014; Caldwell et al., 2024). Finally, a 20G venous catheter was inserted into a peripheral arm vein as the site for sodium lactate infusion.

**Passive hyperlactataemia.** Arterial lactate concentration was increased and clamped at 4 mmol/l and then 8 mmol/l for 10 min at each stage. We used a 1.0 M sodium lactate solution with a pH of 4.7 tested for pyrogenicity and sterility (Macdonald's Prescriptions, Vancouver, BC, Canada). The infusion rate started at 15 mg/kg/min using an infusion pump (Harvard Apparatus PHD Ultra, Holliston, MA, USA). The rate of infusion was continuously adjusted within a range of 10–30 mg/kg/min

to clamp arterial lactate at the desired concentration based on frequent spot-checks of arterial blood (see 'Blood sampling'). Measures (arterial and jugular bulb blood sample, and CBF) were taken when the target concentrations were reached, and after 10 min for each stage. After the last stage, the infusion was stopped and participants rested for 1 h to allow complete washout.

**Active hyperlactataemia.** Participants completed 2–3 1-min efforts on a semi-recumbent cycle ergometer (Lode Angio Imaging, Groningen, the Netherlands) at 80–100% of their maximum work rate ($W_{max}$). Work rate was increased slightly with each effort to ramp circulating lactate in a stepwise manner to target a lactate concentration of 4 mmol/l and then 8 mmol/l. Immediately after confirmation of the desired lactate concentration (via arterial blood sample spot checks), arterial and jugular bulb blood samples were drawn simultaneously alongside measures of CBF. Measurements were obtained with the participants at rest on the semi-recumbent cycle ergometer ~2–3 min after exercise cessation ($2.4 \pm 1.2$ min).

## Blood sampling

At each time point, 1.0 ml each of radial artery and jugular venous blood was simultaneously drawn into pre-heparinized syringes (SafePICO, Radiometer, Copenhagen, Denmark). All samples were analysed immediately for metabolite, blood gas, electrolyte and oximetry values using a commercial blood gas analyser (ABL90 FLEX, Radiometer). This analysis was inclusive of measures of the partial pressures of arterial and venous carbon dioxide ($P_{aCO_2}/P_{vCO_2}$) and oxygen ($P_{aO_2}/P_{vO_2}$), oxygen saturation ($S_{aO_2}/S_{vO_2}$), glucose and lactate concentrations ($C_{aGlc}/C_{vGlc}$ and $C_{aLac}/C_{vLac}$, respectively), hydrogen ion concentration ($[H^+]$), bicarbonate ion concentration ($[HCO_3^-]$), haemoglobin concentration ($[Hb]$), haematocrit ($[HCT]$), and pH.

## Cerebral vascular ultrasound

Simultaneous diameters and velocities of the right internal carotid artery (ICA) and left vertebral artery (VA) were measured using 10 MHz multi-frequency linear array duplex ultrasound (Terasmart uSMart 3300, Teratech, Burlington, MA, USA). Simultaneous B-mode imaging and pulse-wave mode were used to record arterial diameter and beat-by-beat velocity, respectively, and each video file was a 1-min recording. All scans were screen captured and stored at 30 Hz as video files for later offline analysis utilising customised edge detection and wall tracking software (BloodFlow Analysis, Version 5.1) to mitigate observer bias (Woodman et al.,

2001). All analyses were performed with the investigator blinded to the participant, condition and lactate stage. The vessel location was determined on an individual participant basis to allow for the highest-quality image acquisition, with the same location and insonation angle repeated within-participant. All recordings were made in accordance with published technical recommendations (Thomas et al., 2015). Calculations of global cerebral blood flow (gCBF) (eqns 2 and 3) were used to quantify cerebral glucose, lactate and oxygen metabolism as the product of CBF arterial and internal jugular vein concentration differences (eqns 6–9).

### Data analyses

Arterial ($C_{aO_2}$) and venous ($C_{vO_2}$) oxygen contents were calculated as (Severinghaus, 1966):

$$C_{aO_2} \text{ and } C_{vO_2} = [\text{Hb}] \times 1.34 \times [S_{O_2}/100] + 0.003 \times P_{O_2} \quad (1)$$

where 1.34 is the binding capacity of Hb (g/dl) for a given $S_{O_2}$ (%) and 0.003 is the fraction of oxygen dissolved in the blood (Lumb & Thomas, 2020; West & Luks, 2020).

For ultrasound scans, volumetric blood flow ($\dot{Q}$) was calculated from blood flow analysis data using the following formula as previously described (Willie et al., 2012):

$$\dot{Q}_{\text{ICA}} \text{ and } \dot{Q}_{\text{VA}} \text{ (mL/min)} = (0.5 \times \text{peak envelope velocity})$$
$$\times \left(\pi \left(0.5 \times \text{diameter}\right)^2\right) \times 60 \quad (2)$$

Global cerebral blood flow (gCBF) was calculated as twice the sum of unilateral assessments of ICA and VA blood flow:

$$\text{gCBF (mL/min)} = 2 \times \left(\dot{Q}_{\text{ICA}} + \dot{Q}_{\text{VA}}\right) \quad (3)$$

Oxidative glucose and carbohydrate index (OGI, OCI) were calculated from trans-cerebral blood samples to quantify brain carbohydrate uptake relative to oxygen:

$$\text{OGI} = \left(C_{aO_2} - C_{vO_2}\right) / \left(C_{aGlc} - C_{vGlc}\right) \quad (4)$$

$$\text{OCI} = \left(C_{aO_2} - C_{vO_2}\right) /$$
$$\left(C_{aGlc} - C_{vGlc} + [0.5\left(C_{aLac} - C_{vLac}\right)]\right) \quad (5)$$

These ratios indicate the amount of oxidative to non-oxidative carbohydrate metabolism such that a value of 6 indicates complete oxidation of glucose/carbohydrates and a value below 6 indicates excess glucose/carbohydrate uptake and non-oxidative metabolism. The lactate difference is divided by 2 as 1 mole of lactate provides half a glucose equivalent and

thus requires half the amount of oxygen for complete oxidation (Dalsgaard et al., 2004).

The cerebral metabolic rate of oxygen, lactate, glucose and carbohydrate was calculated from trans-cerebral exchange of arterial and venous samples as follows:

$$\text{CMR}_{O_2} \text{ (mL/min)} = (\text{CBF}/100) \times \left(C_{aO_2} - C_{vO_2}\right) \quad (6)$$

$$\text{CMR}_{\text{Glc}} \text{ (mmol/min)} = (\text{CBF}/1000) \times (C_{aGlc} - C_{vGlc}) \quad (7)$$

$$\text{CMR}_{\text{iLac}} \text{ (mmol/min)} = (\text{CBF}/1000) \times (C_{aLac} - C_{vLac}) \quad (8)$$

$$\text{CMR}_{\text{iCho}} \text{ (mmol/min)} = (\text{CBF}/1000) \times [(C_{aGlc} - C_{vGlc})$$
$$+ 0.5 \times (C_{aLac} - C_{vLac})] \quad (9)$$

$\text{CMR}_{\text{iLac}}$ and $\text{CMR}_{\text{iCho}}$ are termed indices for these metabolic rates due to the continuous release of lactate from the brain making it impossible to discern whether a change in the arterial–venous difference is due to a rise in cerebral uptake or a decrease in cerebral production without the use of isotope tracers.

Extraction fractions for oxygen (oxygen extraction fraction; OEF), glucose, lactate and carbohydrate were calculated from trans-cerebral exchange of arterial and venous samples. Although termed lactate extraction (%), this outcome reflects net trans-cerebral lactate concentration differences. This difference is influenced by both lactate uptake and endogenous lactate production from the brain.

$$\text{OEF (\%)} = \left(\left(C_{aO_2} - C_{vO_2}\right)/C_{aO_2}\right) \times 100 \quad (10)$$

$$\text{Glucose extraction (\%)} = \left((C_{aGlc} - C_{vGlc})/C_{aGlc}\right) \times 100 \quad (11)$$

$$\text{Lactate extraction (\%)} = \left((C_{aLac} - C_{vLac})/C_{aLac}\right) \times 100 \quad (12)$$

$$\text{Carbohydrate extraction (\%)} = (C_{aGlc} - C_{vGlc}) + 0.5$$
$$\times (C_{aLac} - C_{vLac})/(C_{aGlc} + 0.5 \times C_{aLac}) \times 100 \quad (13)$$

Cerebral delivery of oxygen ($\text{CD}_{O_2}$), glucose ($\text{CD}_{\text{Glc}}$), lactate ($\text{CD}_{\text{Lac}}$) and carbohydrate were calculated as:

$$\text{CD}_{O_2} \text{ (mL/min)} = C_{aO_2} \times \left(\text{gCBF}/100\right) \quad (14)$$

$$\text{CD}_{\text{Glc}} \text{ (mmol/min)} = C_{aGlc} \times \left(\text{gCBF}/1000\right) \quad (15)$$

$$\text{CD}_{\text{Lac}} \text{ (mmol/min)} = C_{aLac} \times \left(\text{gCBF}/1000\right) \quad (16)$$

$$\text{CD}_{\text{Cho}} \text{ (mmol/min)} = (C_{aGlc} \times 0.5 \times C_{aLac})$$
$$\times \left(\text{gCBF}/1000\right) \quad (17)$$

Haematological responses for the percent change in estimated plasma volume (PV) at 4 mmol/l and 8 mmol/l time points compared to baseline was calculated (Dill & Costill, 1974):

$$\%\Delta PV = 100\left[\left(Hb_{pre}/Hb_{post}\right)\left(100 - Hb_{post}\right)\right/$$
$$\left(100 - Hb_{pre}\right) - 1\right] \qquad (18)$$

Plasma apparent strong ion difference (SID) was calculated as the sum of the strong cation concentrations minus the sum of the strong anion concentrations as:

$$[SID]\left(mEq/l\right) = \left(\left[Na^+\right] + \left[K^+\right] + \left[Ca^{2+}\right] - \left[Cl^-\right]\right.$$
$$\left. + \left[Lac^-\right]\right) \text{ (Stewart, 1983)} \qquad (19)$$

### Statistical analysis

Sample size ($n = 13$) was determined by practical constraints given the invasive nature of the study protocol. Large effect sizes for primary outcomes of interest supported adequate power to detect significant effects (CMR$_{Glc}$, $\eta_p^2 = 0.141$) (Cohen, 1992). All statistical analyses were conducted using SPSS (Version 29; IBM Corp., Armonk, NY, USA). Data were normally distributed as assessed by visual inspection of Q–Q plots and homoscedasticity of the Studentized residuals plotted against the predicted values. Linearity of relationships was established by visual inspection of scatterplots. Data were analysed using a linear mixed model for repeated measures with fixed factors of condition (exercise *vs.* sodium lactate infusion) and lactate stage (baseline, 4 mmol, 8 mmol) on outcome measures. Fixed effects included condition, lactate stage and their interaction. A random intercept and random slope for lactate stage were included to account for individual differences in baseline values and response to lactate concentration. Repeated measures were modelled using a diagonal covariance structure for the repeated factor (condition $\times$ lactate concentration). Significance of statistical analyses was determined at $P < 0.05$, with significant interactions explored using pairwise comparisons with a Bonferroni correction. Figures were generated using GraphPad Prism (v10.3.1, GraphPad Software, Boston, MA, USA).

### Results

Thirteen participants were included in the final analysis, with exercise conditions reported for 10 participants at the 4 mmol lactate stage and 12 participants at the 8 mmol stage for reasons of overshooting target lactate concentrations of ∼4 or ∼8 mmol, respectively. Summary data for all metabolic, cardiovascular and cerebrovascular data during passive infusion and exercise protocols can be seen in Appendix A (Table A1).

As intended, similar increases in arterial lactate were observed with the passive and active conditions ($P = 0.679$, Fig. 1*B*), with both conditions reaching the target lactate concentrations of $4.0 \pm 0.1$ and $8.2 \pm 0.2$ mmol/l ($P < 0.001$ differences between all lactate stages). Arterial glucose concentrations were greater during the exercise condition compared to the passive infusion ($P = 0.0160$), with no influence of circulating lactate concentrations ($P = 0.859$, Fig. 1*A*) and no condition by lactate stage interaction effects present ($P = 0.0990$).

### Cerebral blood flow and control

Differences in gCBF and its primary determinants between infusion *versus* exercise are highlighted in Fig. 2. Significant increases in gCBF occurred during the passive infusion present at 8 mmol/l compared to baseline ($P < 0.001$) 4 mmol/l stage ($P = 0.0110$). No significant differences were present during exercise-hyperlactataemia (baseline *vs.* 4 mmol/l: $P = 0.232$, 4 *vs.* 8 mmol/l: $P = 0.212$), resulting in a greater gCBF during the passive infusion at the 8 mmol/l stage *versus* exercise ($P < 0.001$). A significant condition by stage interaction was present for $P_{aCO_2}$ ($P < 0.001$, Fig. 2*B*), whereby no changes in $P_{aCO_2}$ from baseline occurred during the passive infusion at 4 mmol/l ($P = 0.660$) and 8 mmol/l ($P = 0.283$). However, during exercise, $P_{aCO_2}$ was increased from baseline at 4 mmol/l ($P < 0.001$), and thereafter decreased at the 8 mmol/l stage ($P < 0.001$ *vs.* baseline and 4 mmol/l). As such, a lower $P_{aCO_2}$ was evident during exercise at 8 mmol compared to infusion ($P < 0.0001$), with no group differences at 4 mmol/l ($P = 0.949$). Mean arterial pressure (MAP) was not influenced by hyperlactataemia during either condition ($P = 0.605$, Fig. 2*C*), but was elevated in the exercise condition compared to passive infusion ($P = 0.00300$).

### Cerebral oxygen and substrate delivery, extraction, and metabolism

The delivery and extraction of oxygen and carbohydrates (i.e. lactate and glucose) during both the passive and active conditions are presented in Fig. 3. Interaction effects for CD$_{O_2}$ ($P = 0.0240$) and CD$_{Glc}$ were present ($P < 0.001$), with a greater O$_2$ and glucose delivery occurring at 8 mmol/l during passive infusion *vs.* exercise owing to higher CBF ($P = 0.0100$ and $P < 0.001$, respectively). CD$_{Lac}$ was greater during passive infusion *versus* exercise conditions ($P = 0.0320$), again driven by higher CBF. CD$_{Cho}$ was increased with increasing lactate availability in both active and passive conditions ($P < 0.001$), with differences present between all lactate stages ($P < 0.001$). Interaction effects showed a greater CD$_{Cho}$ in the passive

infusion condition compared to passive infusion at the 8 mmol stage only ($P < 0.001$).

Generally, changes in OEF were reciprocal to corresponding changes in $CD_{O_2}$, with interaction effects for cerebral OEF with increasing lactate between conditions ($P < 0.001$). At 8 mmol/l lactate concentration, OEF was higher during exercise *versus* passive infusion ($P < 0.001$), effectively compensating for decreased $CD_{O_2}$ during exercise. Glucose extraction fraction (EF, %) showed no differences between conditions ($P = 0.791$),

with similar decreases present in both conditions at the 8 mmol stage compared to baseline ($P = 0.005$) and 4 mmol ($P = 0.0100$). Lactate concentration difference across the brain did not differ between conditions ($P = 0.917$) but increased (brain lactate uptake) with progressive hyperlactataemia ($P < 0.001$), both at 4 and at 8 mmol/l compared to baseline ($P < 0.001$ for both), but with no differences between the 4 and 8 mmol stages ($P = 0.0600$).

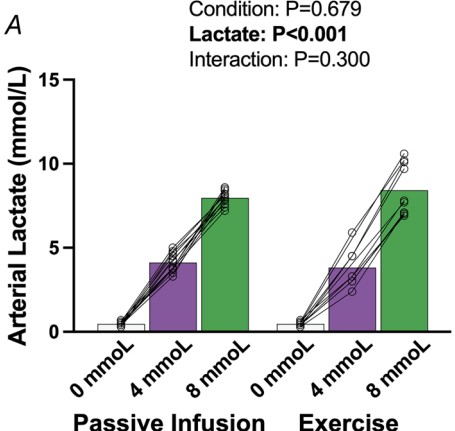
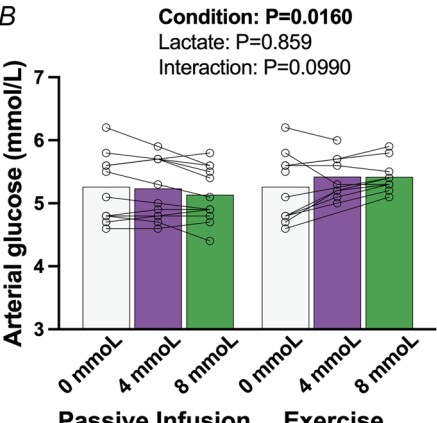

**Figure 1. Arterial glucose and lactate concentrations during targeted matched increases in lactate between passive infusion *versus* exercise**

*A*, lactate; *B*, glucose. Bars indicate the group mean response between conditions and lactate stages, with individual participants shown by symbols. Statistics were performed using a mixed model for repeated measures analysis, with fixed factors of condition (passive infusion, exercise) and lactate concentration (0, 4, 8 mmol). When an interaction is present, *P*-values are presented for the between condition comparisons. Interaction for lactate concentrations within each condition are presented in text. For the passive infusion, *n* = 13 for all conditions (0, 4 and 8 mmol/l), and for the exercise protocol *n* = 13 at 0 mmol/l, *n* = 10 at 4 mmol/l, and *n* = 12 at 8 mmol/l.

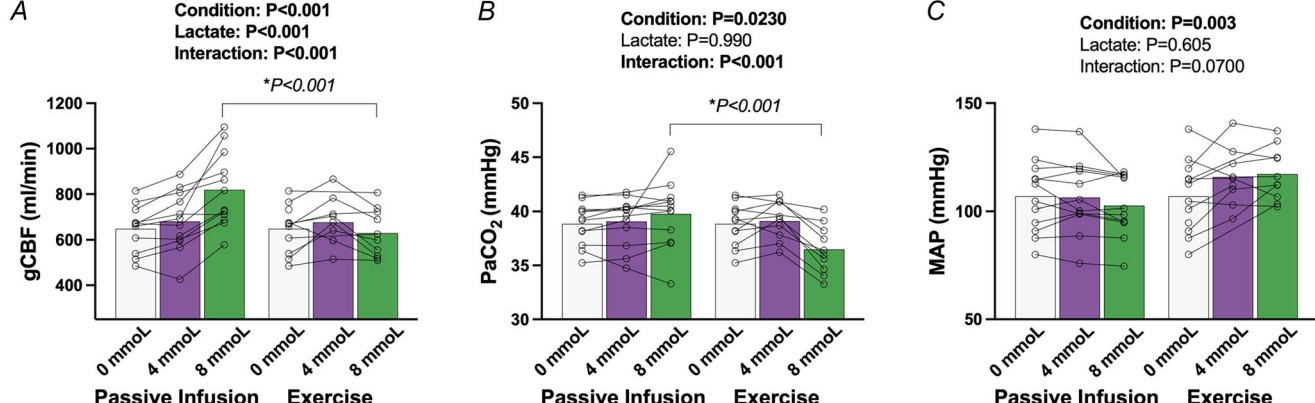

**Figure 2. The response of gCBF and key determinants to matched increases in arterial lactate between passive infusion *vs*. exercise**

*A*, gCBF; *B*, $P_{aCO_2}$; *C*, mean arterial pressure. Bars indicate the group mean response between conditions and concentrations, with individual participants shown by symbols. Statistics were performed using a mixed model for repeated measures analysis, with fixed factors of condition (passive infusion, exercise) and lactate concentration (0, 4, 8 mmol). When interaction is present, *P*-values are presented for the between condition comparisons. Interaction for lactate concentrations within each condition are presented in text. For the passive infusion protocol, *n* = 13 for all conditions (0, 4, and 8 mmol/l), and for the exercise protocol, *n* = 13 at 0 mmol/l, *n* = 10 at 4 mmol/l, and *n* = 12 at 8 mmol/l. gCBF, global cerebral blood flow; $P_{aCO_2}$, arterial carbon dioxide.

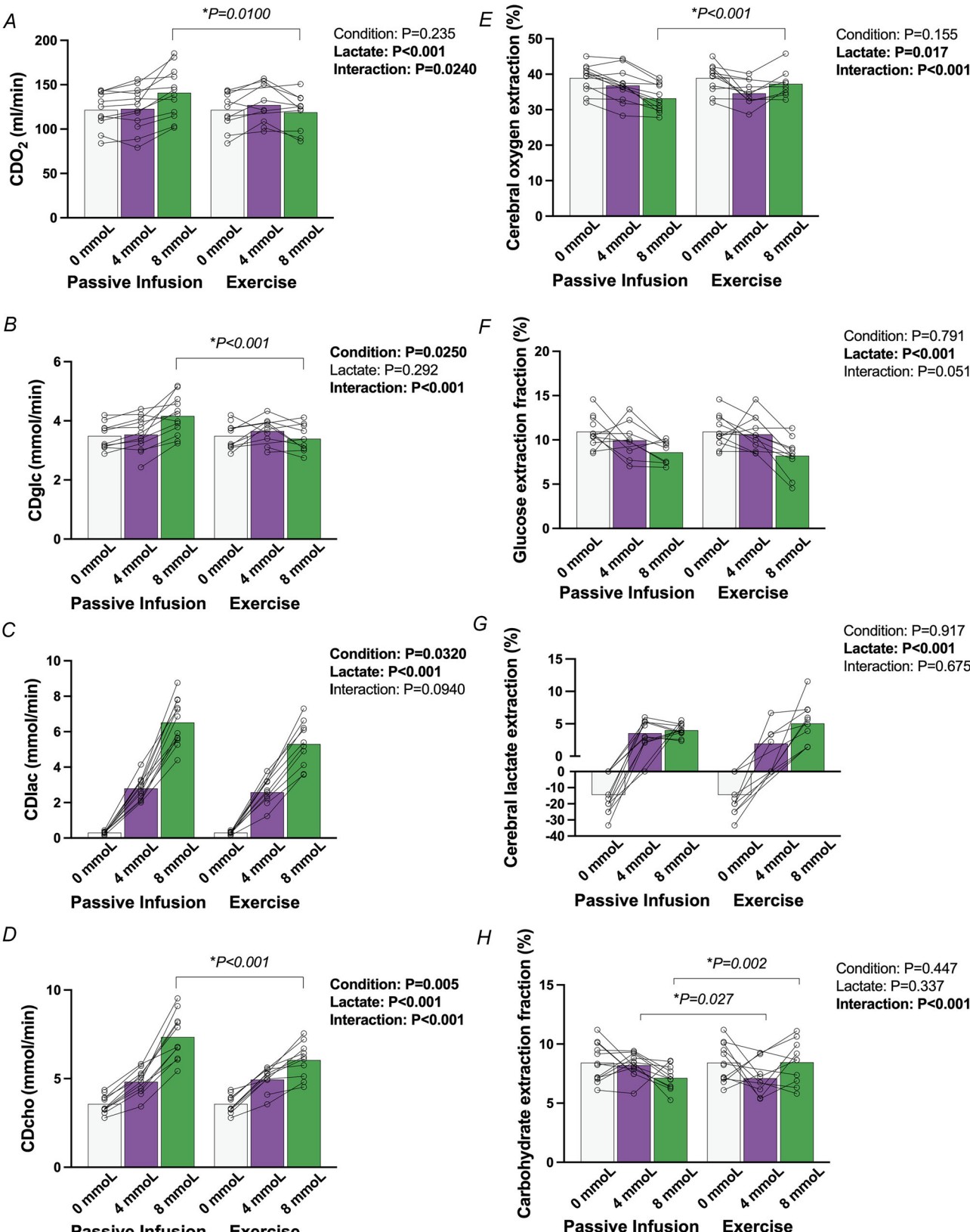

**Figure 3. Cerebral oxygen and carbohydrate delivery and extraction responses during matched increases in circulating lactate during passive infusion compared to exercise**
*A*, CD$_{O_2}$; *B*, CD$_{Glc}$; *C*, CD$_{Lac}$; *D*, CD$_{Cho}$; *E*, OEF; *F*, Glc EF; *G*, Lac EF; *H*, CHO EF. Bars indicate the group mean response between conditions and concentrations, with individual participants shown by symbols. Statistics were

performed using a mixed model for repeated measures analysis, with fixed factors of condition (passive infusion, exercise) and lactate concentration (0, 4, 8 mmol). When interaction is present, *P*-values are presented for the between condition comparisons. Interaction for lactate concentrations within each condition are presented in text. For the passive infusion protocol, *n* = 13 for all conditions (0, 4, and 8 mmol/l), and for the exercise protocol *n* = 13 at 0mmol/l, *n* = 10 at 4 mmol/l, and *n* = 12 at 8 mmol/l. $CD_{Cho}$, cerebral carbohydrate delivery; $CD_{Lac}$, cerebral lactate delivery; $CD_{O_2}$, cerebral oxygen delivery.

$CMR_{O_2}$ did not differ between conditions ($P = 0.901$) and remained unchanged across all lactate stages ($P = 0.610$, Fig. 4*A*). $CMR_{Glc}$ decreased with increasing lactate availability ($P = 0.00900$), with no differences observed between the passive infusion and exercise conditions ($P = 0.373$). Pairwise comparisons showed a reduction in $CMR_{Glc}$ at 8 mmol compared to baseline ($P = 0.0100$, Fig. 4*B*), with no significant differences at 4 mmol ($P = 0.390$). $CMR_{iLac}$ increased significantly with increasing lactate availability ($P < 0.001$), independent of condition ($P = 0.972$, Fig. 4*C*). Pairwise comparisons showed differences were present between all lactate concentrations ($P < 0.001$ for all). The $CMR_{iCho}$ was increased with increasing circulating lactate availability ($P = 0.00700$), with no differences between conditions ($P = 0.106$). OCI decreased with increasing circulating lactate ($P < 0.001$), with reductions observed at both 4 mmol ($P = 0.0100$) and 8 mmol ($P < 0.001$),

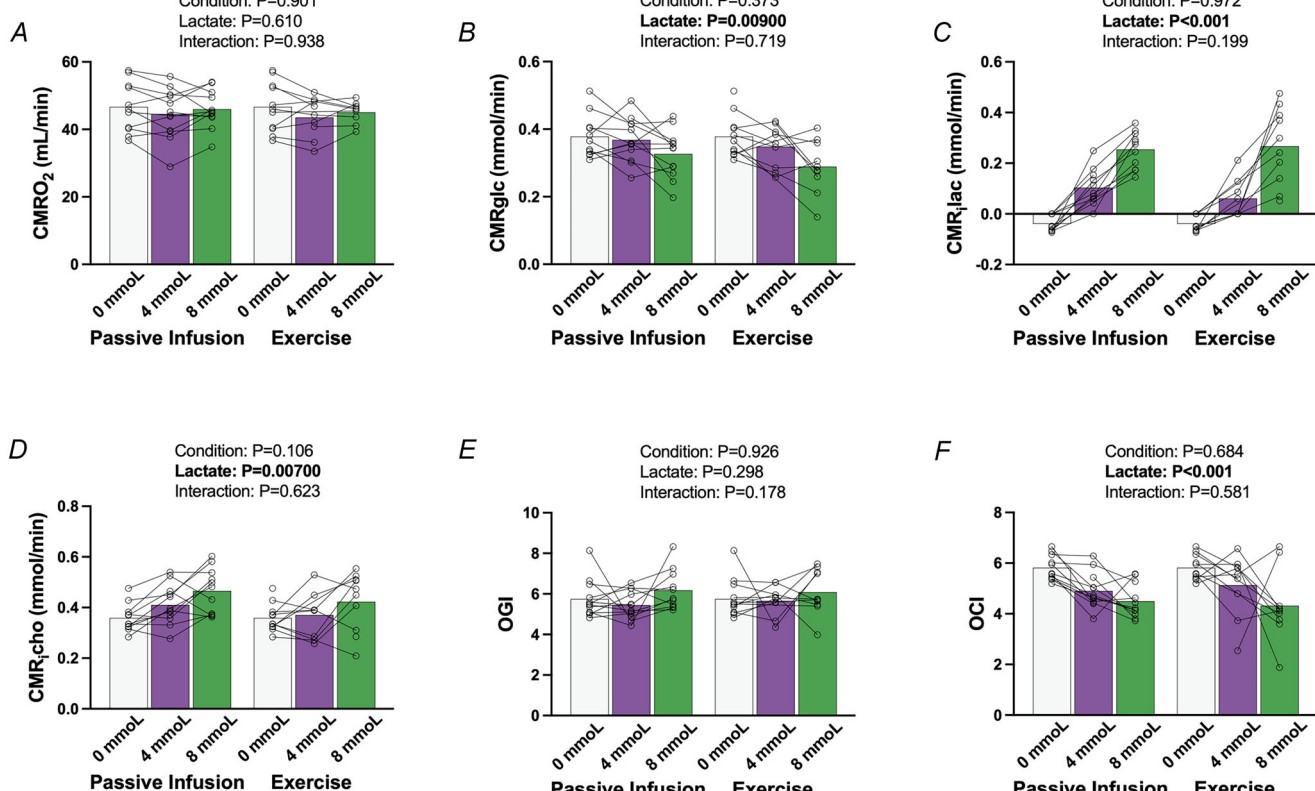

**Figure 4. Cerebral metabolic responses to matched increases in circulating lactate during passive infusion compared to exercise**

*A*, $CMR_{O_2}$; *B*, $CMR_{Glc}$; *C*, $CMR_{iLac}$; *D*, $CMR_{iCho}$; *E*, OGI; *F*, OCI. Bars indicate the group mean response between conditions and concentrations, with individual participants shown by symbols. Statistics were performed using a mixed model for repeated measures analysis, with fixed factors of condition (passive infusion, exercise) and lactate concentration (0, 4, 8 mmol). When interaction is present, *P*-values are presented for the between condition comparisons. Interaction for lactate concentrations within each condition are presented in text. For the passive infusion protocol, *n* = 13 for all conditions (0, 4, and 8 mmol/l), and for the exercise protocol *n* = 13 at 0 mmol/l, *n* = 10 at 4 mmol/l, and *n* = 12 at 8 mmol/l. $CMR_{Glc}$, cerebral metabolic rate of glucose; $CMR_{iCho}$, cerebral metabolic rate of carbohydrate index; $CMR_{iLac}$, cerebral metabolic rate of lactate index; $CMR_{O_2}$, cerebral metabolic rate of oxygen; OCI, oxidative carbohydrate index; OGI, oxidative glucose index.

independent of condition ($P = 0.684$). OGI remained unchanged with increasing lactate availability ($P = 0.298$) and also unaffected by condition ($P = 0.926$).

## Discussion

This study has three primary findings: (1) regardless of endogenous or exogenous increases in lactate, hyperlactataemia causes a dose-dependent decrease in $CMR_{Glc}$; (2) lactate metabolism preferentially sustains $CMR_{O_2}$, and contributes up to 24% of cerebral fuel provision at 8 mmol/l circulating concentration; and (3) the vast surplus of carbohydrate delivered to the brain during hyperlactataemia increased brain carbohydrate uptake, but not oxidation, providing evidence of a dose-dependent increase in non-oxidative glucose metabolism. The similar decrease in $CMR_{Glc}$ matched increases in circulating lactate in two strikingly different physicochemical and physiological contexts provides further evidence that lactate is the preferential fuel source for the brain, sparing glucose for its energetic requirements.

### Lactate: a glucose-sparing oxidative substrate for cerebral energetics

Our findings align with previous research showing that cerebral lactate metabolism increases in direct proportion to its availability in circulation, regardless of whether it is increased via exogenous (Boumezbeur et al., 2010; Smith et al., 2003) or endogenous sources (Ide et al., 1999; Siebenmann et al., 2021; Van Hall et al., 2009). Previous data using FDG-PET have similarly shown $CMR_{Glc}$ decreases during and after exercise when circulating lactate is elevated (Green et al., 2025; Kemppainen et al., 2005). In these FDG-PET studies, neither cerebral lactate nor oxygen metabolism was measured (Green et al., 2025; Kemppainen et al., 2005), but it is presumed that the decrease in $CMR_{Glc}$ is met with a compensatory increase in lactate metabolism to sustain cerebral energetics. However, without direct cerebral arteriovenous sampling, these interpretations remained to be substantiated. For the first time, the present study provides this evidence, confirming that $CMR_{Glc}$ is reduced in a dose-dependent manner with progressive increases in circulating lactate and that $CMR_{O_2}$ is sustained in the face of declining $CMR_{Glc}$. Compensatory brain lactate metabolism is consistent with the observation that $CMR_{Glc}$ is reduced by a similar amount at matched lactate concentrations despite dramatically different physicochemical environments between the two conditions. For example, lactate infusion increased gCBF by 27% and thus significantly augmented lactate, glucose and oxygen delivery to the brain compared to exercise (Fig. 3). Despite these marked elevations in cerebral fuel delivery, the decline in $CMR_{Glc}$ was remarkably similar between conditions (Fig. 4).

### Elevated lactate provides a surplus of usable fuel for the brain, but does not increase $CMR_{O_2}$: evidence for an upregulation of aerobic glycolysis

The present data show that when lactate is made available in the circulation, there is a 70–100% increase in carbohydrate delivery to the brain (Fig. 3). Despite this abundance of usable fuel being delivered to the brain, we observed no increase in $CMR_{O_2}$, indicating unchanged total energy turnover. Thus, in the conditions of the current study, providing a surplus of carbohydrate and oxygen does not translate into increased energy metabolism. However, the dose-dependent *decrease* in OCI and *increase* in $CMR_{iCho}$ with increasing lactate concentrations indicates that cerebral non-oxidative glucose metabolism is upregulated, that is, a greater proportion of carbohydrate is being extracted by the brain but not oxidized. These findings support the concept of metabolic flexibility, wherein the brain utilizes lactate for energy production to spare glucose for alternative pathways via carbon offshoots from glycolysis. These glucose-derived glycolysis intermediates may subserve alternative fates beyond that of energy production via conversion to pyruvate or lactate (Benveniste et al., 2018; Dienel, 2019). Although the present human experimental model cannot directly address the fate of this excess glucose, several potential pathways warrant discussion. These include diversion into the pentose phosphate pathway (PPP), which represents a relatively slow biosynthetic pathway in managing oxidative stress (Baquer et al., 1988; Hothersall et al., 1982), serving as a precursor for several structural compounds (Jones et al., 1975; Ramsey et al., 1971), storage via the glycogen shunt (Dienel, 2019; Matsui, 2021; Matsui et al., 2012; Shulman et al., 2001) or dispersal through perivascular–lymphatic clearance (Dienel & Lauritzen, 2025). We propose the most likely explanation is that lactate oxidation spares glucose for PPP and storage as glycogen (Dienel et al., 2002), while simultaneously some lactate is released from the brain via perivascular drainage (Ball et al., 2010). This glucose sparing effect may be reflective of lactate's more constrained metabolic fate.

In contrast to glucose, lactate follows a single metabolic pathway as a substrate for oxidative phosphorylation. As such, lactate does not enter glycolysis and it can be assumed that lactate taken up by the brain is oxidized to sustain $CMR_{O_2}$ (Dalsgaard et al., 2004; Overgaard et al., 2012; Van Hall et al., 2009). Thus, at least based on pre-clinical studies, this surplus of lactate consumed by the brain is allowing glucose to fulfil non-energetic requirements related to stress resistance (Baquer et al.,

1988; Hothersall et al., 1982), growth (Jones et al., 1975; Ramsey et al., 1971) and energy storage (Dienel, 2019; Matsui, 2021; Matsui et al., 2012). Some *in vitro* work also indicates that lactate is preferentially used specifically during neuronal activation, potentially due to the lower ATP costs associated with glycolysis, and in sparing glucose (Schurr & Gozal, 2011; Serres et al., 2003), providing evidence that lactate can also fuel neuronal activity during heightened energetic demand. Furthermore, the increased expression of MCTs under conditions of elevated lactate availability further supports the hypothesis that substrate preference may shift from glucose to lactate when lactate concentrations exceed a certain threshold, even in the absence of metabolic stress (Pérez-Escuredo et al., 2016; Smith et al., 2003).

### Reconciling the role of substrate delivery *versus* demand in driving changes in cerebral glucose and lactate metabolism

A key consideration in regards to cerebral substrate metabolism is the complex interplay between delivery and extraction of various substrates, and the preference of cerebral metabolism for certain metabolic pathways. Despite a greater cerebral delivery of fuel and oxygen during the passive lactate infusion, similar shifts towards greater $CMR_{iLac}$ and lower $CMR_{Glc}$ were evident in both protocols. Furthermore, total cerebral energy turnover, that is, $CMR_{O_2}$, was unaffected by these strikingly different rates of cerebral substrate (and oxygen) delivery. This observation reinforces the notion that increased substrate delivery does not necessarily drive metabolic flux; rather it is dictated by metabolic demands, which were not different between passive infusion and exercise (Fig. 4). This aligns with the lower basal $CMR_{Glc}$ evident in individuals with type II diabetes, whose brains are chronically inundated with glucose, but are unable or unwilling to extract and metabolize it (Deery et al., 2024; Hwang et al., 2017; Jensen et al., 2025). That there was no increase in cerebral energetics with a surplus of fuel delivery via lactate indicates that indiscriminately flooding the brain with fuel may alter substrate choice, but will not force an increase in total energy turnover in healthy humans. This may differ in disease states where $CMR_{Glc}$ is attenuated, and providing alternative cerebral metabolic substrates like ketone bodies appears to improve net energy turnover as indicated by no decrease in $CMR_{Glc}$ (Cunnane et al., 2016; Fortier et al., 2019). Thus, the absence of a decline in $CMR_{Glc}$ in response to increased lactate or ketones may indicate cerebral energetic failure in some disease states (Green et al., 2025).

### Lactate as a powerful cerebral vasodilator?

In the absence of disease or trauma, circulating lactate is only elevated substantially following intense exercise, which also causes hypocapnia. Hypocapnia is the most potent physiological cerebral vasoconstrictor and is largely responsible for the cerebral hypoperfusion that occurs during intense exercise (Lassen, 1959; Willie et al., 2014). This confounding influence of hypocapnia makes it difficult to isolate a role of exercise-induced hyperlactataemia on cerebrovascular tone. However, we observed a 27% increase in gCBF when circulating lactate was passively increased to 8 mmol/l (Fig. 2). This increase in CBF is beyond what would be expected by mild hypercapnia (+1 mmHg) (Willie et al., 2012) and moderate haemodilution ($\sim$8–9%) (Hoiland et al., 2016). It is therefore plausible that lactate *per se* is a cerebral vasodilator, with its actions in part masked by the opposing powerful vasoconstriction of hypocapnia during intense exercise. Animal data suggest that elevations in lactate may alter vessel tone via direct hyperpolarization of vascular smooth muscle (Barron & Nair, 2003; Hester et al., 1980) and modulation of prostaglandin $E_2$ availability (Gordon et al., 2008) promoting vascular smooth muscle-dependent vasodilation. While these findings indicate a potential vasodilatory role for lactate, direct evidence in humans is lacking and remains limited to evidence of increases in CBF rather than evidence on cerebral vessel tone *per se*. Further, we cannot isolate the role of lactate from a potential vasodilatory role of elevated $HCO_3^-$ (Caldwell, Howe, et al., 2021).

### Experimental considerations

The infusion of sodium lactate resulted in significant haemodilution, hypernatraemia and metabolic alkalosis (Appendix Fig. A1). The increase in arterial plasma pH is a result of infusing hypertonic sodium lactate solution, which results in excess extracellular $Na^+$ concentration leading to shifts in hydrogen and bicarbonate concentrations via its effects on SID (Miller et al., 2005). However, previous work has demonstrated that decreases in extracellular pH can improve cerebral lactate uptake (Knudsen et al., 1991), suggesting that in the present data the magnitude of lactate metabolism may be underestimated with passive increases in arterial lactate.

Another limitation to interpretation of the present study derives from the difference in cerebral lactate turnover with exercise compared to the resting state. While the brain is continuously releasing lactate, exercise is known to accelerate lactate turnover and clearance

rates, such that the rate of cerebral lactate production is known to be higher during exercise (Van Hall et al., 2009). Thus, despite matched arterial lactate concentrations, this discrepancy would result in an underestimation of cerebral lactate and carbohydrate metabolism following exercise when conducting cross-brain blood sampling in the absence of stable isotope tracers. For these reasons, we have represented $CMR_{iLac}$ and $CMR_{iCho}$ as indices since they are likely valid only when cerebral lactate production is fixed.

Cerebral metabolic rates during the exercise condition were measured 2–3 min after each cycling effort when accurate blood flow measurements via ultrasound could be reliably acquired. Exercise has been shown to increase $CMR_{O_2}$ (Smith et al., 2014), although this is not universal (Madsen et al., 1993; Secher et al., 2008; Trangmar et al., 2015), and likely dependent on the mode and intensity of exercise (Smith et al., 2014; Volianitis et al., 2008, 2011). The absence of a change in $CMR_{O_2}$ in response to exercise may be caused by the short duration of the exercise (1 min) and non-steady-state conditions of the measurement, a necessity when the goal is to increase circulating lactate. Nevertheless, the progressive decrease in OCI during exercise is reflective of exercise-induced cerebral activation (Dalsgaard et al., 2002, 2004; Madsen et al., 1993; Rasmussen et al., 2011; Volianitis et al., 2008).

disruption, altered MCT expression and mitochondrial dysfunction (Bergersen, 2015; He et al., 2025). Elucidating these factors may inform targeted strategies, such as modulation of MCT expression or mitochondrial capacity to optimize lactate utilization in pathological conditions. Beyond physiological contexts, data suggest that lactate supports neuronal excitability and plasticity, with potential implications for exercise-related cognitive benefits (Hashimoto et al., 2018; Hayek et al., 2019; Magistretti & Allaman, 2018). Thus, future investigations discerning the interplay between shifts in substrate metabolism and cognitive performance under different exercise conditions presents an important avenue for understanding the implications of exercise on brain health and disease prevention.

## Conclusions

Collectively, the present findings show that when lactate is made available in circulation – whether through passive infusion or exercise – the healthy human brain preferentially metabolizes it over glucose to sustain energetic needs. This sparing effect of lactate on glucose appears to permit glucose to fulfil non-energetic requirements of the brain.

## Implications

The role of lactate as a glucose-sparing energy substate may have potential relevance for populations with impaired glucose metabolism, such as individuals with brain injury (Vespa et al., 2025), neurodegenerative diseases (Goyal et al., 2023) or insulin resistance (Willette et al., 2015), where enhanced lactate metabolism may serve as a neuroprotective mechanism (Berthet et al., 2009; Patet et al., 2016; Won et al., 2012). Of particular relevance to Alzheimer's disease, a loss of non-oxidative glucose metabolism during ageing, i.e. aerobic glycolysis, is strongly associated with cognitive impairment, whereas a preservation of aerobic glycolysis provides protection against Alzheimer's pathology (Goyal et al., 2017, 2023). Here we show lactate increases brain aerobic glycolysis in a dose-dependent manner and thus could provide a metabolic explanation for the neuroprotective effects of exercise (Hayes et al., 2014; Hörder et al., 2018; Wanigatunga et al., 2025). Further investigations into the regulatory mechanisms of lactate uptake and its preferential use over glucose could provide valuable insights into cerebral metabolic flexibility and its role in health and disease. An important consideration for future investigations into the therapeutic potential of lactate supplementation is whether its efficacy is maintained in neurological disease states characterized by BBB

## Appendix A

During infusion, arterial sodium ($Na^+$) concentration was increased at all stages ($P < 0.001$). Increases in arterial $[Na^+]$ also occurred during exercise hyperlactataemia ($P < 0.001$), but to a smaller extent *versus* the infusion (+2.2 mmol *vs.* +5.8 mmol at 8 mmol/l, respectively; $P < 0.001$). During the lactate infusion, [SID] increased progressively across all stages ($P = 0.007$). In contrast, during exercise hyperlactataemia, [SID] decreased from baseline, with significant reductions at 8 mmol/l ($P < 0.001$). Consequently, at 8 mmol/l lactate, [SID] was significantly higher in the infusion condition compared to exercise ($P < 0.001$). Lactate infusion caused progressive alkalaemia ($P < 0.001$), while exercise caused acidaemia at 8 mmol/l only ($P < 0.001$). During lactate infusion, $[HCO_3^-]$ increased across all lactate stages ($P < 0.001$). In contrast, with exercise hyperlactataemia $[HCO_3^-]$ was only significantly elevated at the 8 mmol lactate time-point compared to 4 mmol and baseline ($P < 0.002$). As such, at the 8 mmol stage, $[HCO_3^-]$ was significantly higher with lactate infusion compared to exercise ($P < 0.001$). During lactate infusion, PV expanded and was higher at both 4 mmol and 8 mmol lactate stage *versus* exercise ($P < 0.001$). During exercise, PV contracted and this was significantly lower than baseline at the 8 mmol stage ($P = 0.0250$).

**Table A1. Summary of metabolic, cardiovascular, and cerebrovascular data during passive infusion and exercise induced increases in arterial lactate concentrations**

| Outcome | Baseline 0 mmol | Passive 4 mmol | Passive 8 mmol | Active 4 mmol | Active 8 mmol | P Condition | P Lactate | P Interaction |
|---|---|---|---|---|---|---|---|---|
| Glucose (mmol/l) | 5.3 ± 0.1 | 5.2 ± 0.1 | 5.1 ± 0.1 | 5.4 ± 0.1 | 5.5 ± 0.1 | **0.016** | 0.859 | 0.0990 |
| Lactate (mmol/l) | 0.5 ± 0.1 | 4.1 ± 0.2 | 8.0 ± 0.1 | 3.8 ± 0.4 | 8.4 ± 0.5 | 0.679 | **<0.001** | 0.300 |
| HR (bpm) | 61 ± 4 | 63 ± 3 | 76 ± 8 | 75 ± 4 | 98 ± 8 | **<0.001** | **<0.001** | **<0.001** |
| MAP (mmHg) | 107 ± 5 | 106 ± 5 | 103 ± 4 | 113 ± 5 | 116 ± 5 | **0.00300** | 0.605 | 0.0700 |
| $P_{aCO_2}$ (mmHg) | 38.9 ± 0.6 | 39.1 ± 0.6 | 39.7 ± 0.9 | 39.7 ± 0.7 | 36.9 ± 0.7 | **0.0230** | 0.990 | **<0.001** |
| $P_{aO_2}$ (mmHg) | 96.2 ± 2.0 | 92.9 ± 1.9 | 89.7 ± 2.7 | 103.4 ± 2.2 | 113.2 ± 2.3 | **<0.001** | 0.313 | **<0.001** |
| pH | 7.41 ± 0.01 | 7.43 ± 0.01 | 7.47 ± 0.01 | 7.43 ± 0.01 | 7.38 ± 0.01 | **<0.001** | **<0.001** | **<0.001** |
| Haematocrit (%) | 42.5 ± 1.1 | 41.5 ± 1.1 | 39.8 ± 1.1 | 42.9 ± 1.1 | 43.4 ± 1.1 | **0.002** | 0.647 | **0.0150** |
| $Na^+$ (mmol/l) | 140.2 ± 0.2 | 142.9 ± 0.3 | 145.9 ± 0.5 | 141.8 ± 0.2 | 142.4 ± 0.3 | **<0.001** | **<0.001** | **<0.001** |
| $C_{aO_2}$ (ml/dl) | 18.6 ± 0.5 | 18.1 ± 0.5 | 17.3 ± 0.7 | 18.8 ± 0.6 | 19.2 ± 0.7 | **<0.001** | 0.742 | **0.004** |
| $[HCO_3^-]$ (mmEq l) | 24.4 ± 0.3 | 25.5 ± 0.2 | 28.1 ± 0.2 | 25.8 ± 0.6 | 21.3 ± 0.5 | **<0.001** | **0.006** | **<0.001** |
| SID | 37.2 ± 0.4 | 38.6 ± 0.4 | 41.2 ± 0.6 | 38.8 ± 0.6 | 34.7 ± 0.8 | 0.111 | **0.00700** | **<0.001** |
| ΔPV% | 0 | 3.6 ± 0.3 | 8.9 ± 0.8 | −0.4 ± 0.9 | −2.7 ± 1.1 | **<0.001** | 0.164 | **<0.001** |
| gCBF (ml/min) | 641 ± 32 | 681 ± 33 | 818 ± 43 | 703 ± 36 | 642 ± 36 | **<0.001** | **<0.001** | **<0.001** |
| $CD_{O_2}$ (ml/min) | 119 ± 6 | 123 ± 7 | 141 ± 8 | 132 ± 7 | 122 ± 7 | 0.235 | **0.001** | **0.0240** |
| OCI | 6.1 ± 0.2 | 4.9 ± 0.2 | 5.3 ± 0.8 | 5.1 ± 0.6 | 4.4 ± 0.4 | 0.684 | **0.001** | 0.581 |
| OGI | 5.8 ± 0.2 | 5.82 ± 0.3 | 6.5 ± 0.4 | 5.7 ± 0.3 | 6.1 ± 0.4 | 0.926 | 0.298 | 0.178 |
| OEF (%) | 38.5 ± 1.1 | 36.8 ± 1.2 | 33.2 ± 1.1 | 34.2 ± 1.6 | 38.3 ± 1.5 | 0.155 | **0.0170** | **<0.001** |
| Glucose EF (%) | 10.9 ± 0.4 | 10.5 ± 0.5 | 7.9 ± 0.5 | 9.6 ± 0.7 | 8.3 ± 0.9 | 0.791 | **<0.001** | 0.0510 |
| Lactate EF (%) | −13.5 ± 2.4 | 2.8 ± 0.9 | 3.6 ± 0.5 | 5.1 ± 3.2 | 5.1 ± 1.0 | 0.917 | **<0.001** | 0.675 |
| CHO EF (%) | 8.4 ± 0.4 | 8.2 ± 0.5 | 7.1 ± 0.9 | 7.1 ± 0.6 | 8.7 ± 1.0 | 0.447 | 0.337 | **<0.001** |
| $CMR_{O_2}$ (ml/min) | 46.4 ± 1.9 | 44.6 ± 1.9 | 46.0 ± 2.0 | 44.8 ± 2.0 | 45.8 ± 1.9 | 0.901 | 0.610 | 0.938 |
| $CMR_{Glc}$ (mmol/min) | 0.38 ± 0.02 | 0.37 ± 0.02 | 0.33 ± 0.03 | 0.36 ± 0.02 | 0.33 ± 0.03 | 0.373 | **0.00900** | 0.719 |
| $CMR_{iLac}$ (mmol/min) | −0.04 ± 0.01 | 0.10 ± 0.03 | 0.26 ± 0.03 | 0.06 ± 0.03 | 0.29 ± 0.05 | 0.972 | **<0.001** | 0.199 |
| $CMR_{iCho}$ (mmol/min) | 0.36 ± 0.03 | 0.41 ± 0.05 | 0.47 ± 0.03 | 0.38 ± 0.05 | 0.43 ± 0.03 | 0.106 | **0.00700** | 0.623 |
| $CD_{Cho}$ (mmol/min) | 3.5 ± 0.2 | 4.9 ± 0.2 | 7.4 ± 0.4 | 5.1 ± 0.2 | 6.2 ± 0.3 | **0.00500** | **<0.001** | **<0.001** |
| $CD_{Glc}$ (mmol/min) | 3.4 ± 0.1 | 3.5 ± 0.2 | 4.2 ± 0.2 | 3.7 ± 0.2 | 3.5 ± 0.2 | **0.0250** | 0.292 | **<0.001** |
| $CD_{Lac}$ (mmol/min) | 0.3 ± 0.1 | 2.8 ± 0.2 | 6.5 ± 0.4 | 2.6 ± 0.2 | 5.3 ± 0.4 | **0.0320** | **<0.001** | 0.0940 |

Data presented as means ± SE. Statistics performed using a mixed model for repeated measures analysis, with fixed factors of condition (passive infusion, exercise) and lactate concentration (0, 4, 8 mmol), and any condition × concentration interaction effects. *P*-values shown in bold indicate statistical significance. For the passive infusion protocol, $n = 13$ for all conditions (0, 4, and 8 mmol/l), and for the exercise protocol $n = 13$ at 0 mmol/l, $n = 10$ at 4 mmol/l, and $n = 12$ at 8 mmol/l. $C_{aO_2}$, arterial oxygen content; $CD_{O_2}$, cerebral oxygen delivery; $CMR_{Glc}$, cerebral metabolic rate of glucose; $CMR_{iCho}$, cerebral metabolic rate of carbohydrate index; $CMR_{iLac}$, cerebral metabolic rate of lactate index; $CMR_{O_2}$, cerebral metabolic rate of oxygen; EF, extraction fraction; gCBF, global cerebral blood flow; $HCO_3^-$, bicarbonate; HR, heart rate; MAP, mean arterial pressure; OCI, oxidative carbohydrate index; OEF, oxygen extraction fraction; OGI, oxidative glucose index; $P_{aCO_2}$, arterial carbon dioxide; $P_{aO_2}$, arterial oxygen; glucose EF, glucose extraction fraction.

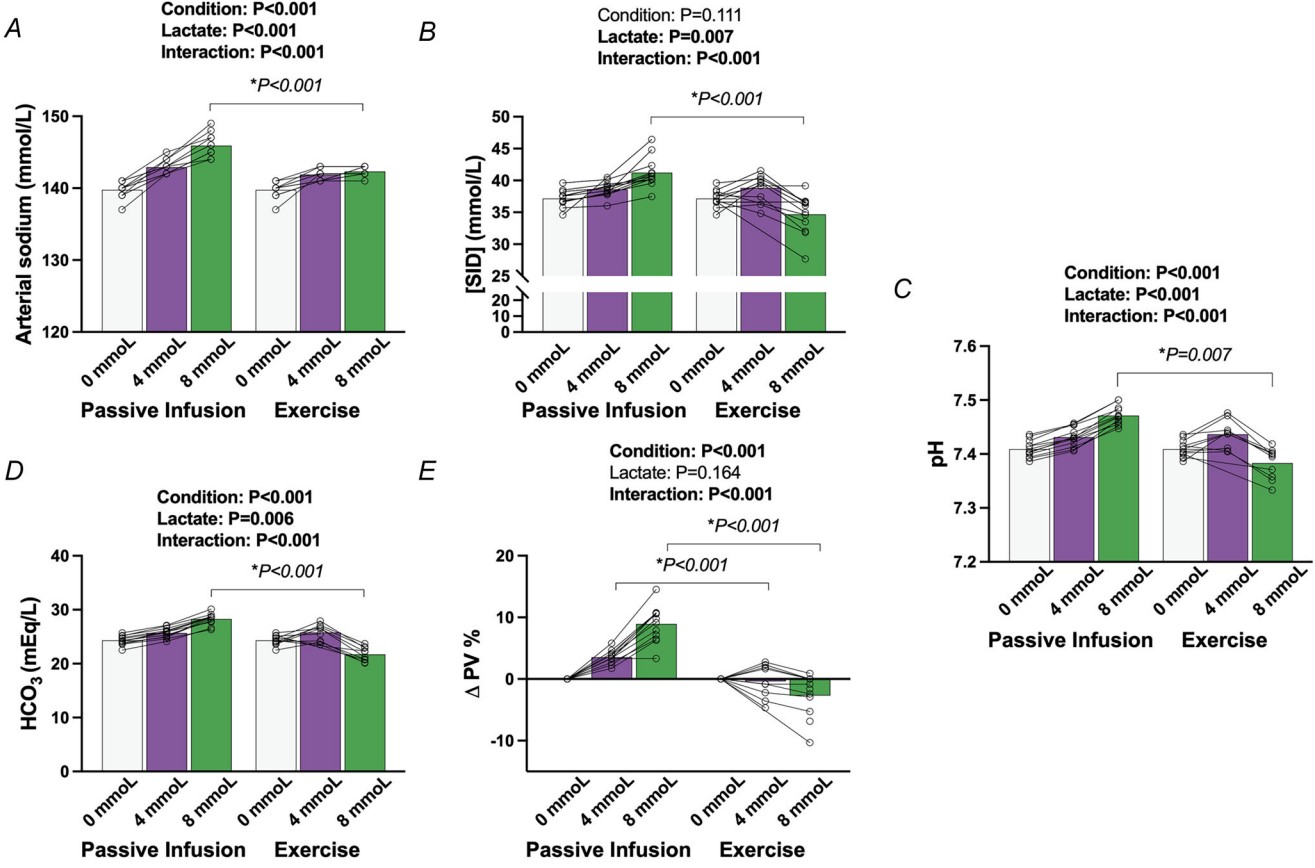

**Figure A1. Physiochemical responses to matched increases in circulating lactate during passive infusion compared to exercise**

*A*, arterial sodium; *B*, arterial [SID]; *C*, arterial pH; *D*, arterial [$HCO_3^-$]; *E*, arterial Δ PV%. Bars indicate the group mean response between conditions and lactate stages, with individual participants shown by symbols. Statistics were performed using a mixed model for repeated measures analysis, with fixed factors of condition (passive infusion, exercise) and lactate concentration (0, 4, 8 mmol). When an interaction is present, *P*-values are presented for the between condition comparisons. Interaction for lactate concentrations within each condition are presented in text. For the passive infusion protocol, $n = 13$ for all conditions (0, 4, and 8 mmol/l), and for the exercise protocol $n = 13$ at 0 mmol/l, $n = 10$ at 4 mmol/l, and $n = 12$ at 8 mmol/l. $HCO_3^-$, bicarbonate; PV, plasma volume; SID, strong ion difference.

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

## Additional information

### Data availability statement

Original data arising from this research are available directly from Dr. Travis D. Gibbons upon reasonable request.

### Competing interests

The authors declare no competing interests.

### Author contributions

T.D.G. and P.N.A. contributed to the conception and design of the study. J.L.K., T.D.G., J.S.D., J.M.J.R.C., M.L.B., J.D.B., J.A.M., T.D.R.M., H.I., A.R.S., C.A.H. and D.B.M. were involved in acquisition, analysis, or interpretation of the data. J.L.K. and T.D.G. drafted the manuscript, and all authors were involved in revising it critically for important intellectual content. All authors have read and approved the final version of this manuscript and agree to be accountable for all aspects of the work in ensuring that questions related to the accuracy or integrity of any part of the work are appropriately investigated and resolved. All persons designated as authors qualify for authorship, and all those who qualify for authorship are listed.

### Funding

This study was funded by a Stober Foundation Postdoctoral Fellowship (J.L.K.), a Canada Research Chair and Natural Sciences and Engineering Research Council of Canada Discovery Grant (P.N.A.).

### Keywords

cerebral lactate metabolism, cerebral substrate switch, endogenous lactate, exogenous lactate

## Supporting information

Additional supporting information can be found online in the Supporting Information section at the end of the HTML view of the article. Supporting information files available:

**Peer Review History**

### Translational perspective

This study tested whether the human brain preferentially uses lactate over glucose as fuel when circulating lactate levels are elevated. Using two different methods – vigorous exercise and passive lactate infusion – we demonstrated that increased lactate availability causes the brain to reduce its glucose consumption by up to 60%, indicating that lactate becomes the preferred energy source. This glucose-sparing effect occurred regardless of whether lactate was elevated through exercise or passive infusion, suggesting a robust and metabolic preference of the brain for lactate under conditions of elevated supply. Despite increased fuel availability, brain oxygen uptake remained unchanged, suggesting that the preserved glucose is directed toward non-oxidative metabolic pathways that support critical brain functions including stress resistance, cellular repair and energy storage. These findings offer important insight for neurological conditions where glucose metabolism is impaired, such as Alzheimer's disease, traumatic brain injury and diabetes-related cognitive decline (Baker et al., 2011; Chen & Zhong, 2013; Costantini et al., 2008). Since lactate can effectively substitute glucose for brain energetic needs, therapeutic strategies that safely elevate lactate levels, whether through controlled exercise, or pharmacological approaches, could maintain brain energy supply when glucose metabolism is compromised. While disease-specific metabolic responses may differ, the brain's apparent metabolic flexibility to use alternative fuels could partly explain why exercise provides cognitive benefits and neuroprotection. Future research should investigate optimal lactate dosing strategies, explore lactate-based therapies for neurodegenerative diseases, and determine whether lactate supplementation can preserve cognitive function in at-risk populations, potentially opening new avenues for treating brain metabolic disorders.

