## [Peer Review History · The Journal of Physiology]

Preferential lactate metabolism in the human brain during exogenous and endogenous hyperlactatemia

Jodie Lauren Koep, Jennifer Duffy, Jay M.J.R. Carr, L. Madden Brewster, Jordan Douglas Bird, Justin A Monteleone, Tenasia Monaghan, Hashim Islam, Andrew Robert Steele, Connor A Howe, David B MacLeod, Philip N Ainslie, and Travis D Gibbons

DOI: 10.1113/JP289216

Corresponding author(s): Jodie Koep (jodie.koep@ubc.ca)

Review Timeline:

Submission Date:	05-May-2025
Editorial Decision:	28-Jul-2025
Revision Received:	11-Aug-2025
Editorial Decision:	02-Sep-2025
Revision Received:	02-Sep-2025
Accepted:	04-Sep-2025

Senior Editor: Harold Schultz

Reviewing Editor: Harold Schultz

Transaction Report:

Dear Dr Koep,

Re: JP-RP-2025-289216 "**Preferential lactate metabolism in the human brain during exogenous and endogenous hyperlactaemia**" by Jodie Lauren Koep, Jennifer Duffy, Jay M.J.R. Carr, L. Madden Brewster, Jordan Douglas Bird, Justin A Monteleone, Tenasia Monaghan, Hashim Islam, Andrew R Steele, Connor A Howe, David B MacLeod, Philip N Ainslie, and Travis D Gibbons

Thank you for submitting your manuscript to The Journal of Physiology. It has been assessed by a Reviewing Editor and by an expert referee and we are pleased to tell you that it is potentially acceptable for publication following satisfactory major revision.

REVISION CHECKLIST:

We look forward to receiving your revised submission.

Yours sincerely,

Harold Schultz
Senior Editor
The Journal of Physiology

REQUIRED ITEMS

- 1) Author photo and profile. First or joint first authors are asked to provide a short biography (no more than 100 words for one author or 150 words in total for joint first authors) and a portrait photograph. These should be uploaded and clearly labelled together in a Word document with the revised version of the manuscript. See Information for Authors for further details.
- 2) The reference list must be in alphabetical order, rather than numbered, to comply with our Journal format.
- 3) Please upload separate high-quality figure files via the submission form.
- 4) Please ensure that any tables are editable and in Word format, and wherever possible, embedded in the article file itself.
- 5) Your paper contains Supporting Information of a type that we no longer publish, including supplementary tables and figures. Any information essential to an understanding of the paper must be included as part of the main manuscript and figures. The only Supporting Information that we publish are video and audio, 3D structures, program codes and large data files. Your revised paper will be returned to you if it does not adhere to our Supporting Information Guidelines.
- 6) We invite you to include a Translational Perspective paragraph in your manuscript. This should be included in the main body of the manuscript after the Acknowledgements. It should describe the wider translational implications of the work, in plain English, for a broad scientific audience. Please use the following guidelines to prepare a Translational Perspective of your paper: https://jp.msubmit.net/cgi-bin/main.plex?form_type=display_requirements#authortranspersp. The Translational Perspective should not exceed 250 words in total and should be presented as a single paragraph. Abbreviations and technical terms must be defined as briefly and simply as possible the first time they are used, unless they are generally/easily understood, e.g. ECG, HIV/AIDS, K⁺ channel. Use language that can be understood by scientists or clinicians with a general knowledge of the topic addressed. Ensure the paragraph includes the hypothesis tested in the paper and accurately reflects the findings of the paper and the implications for future research. Please state the word count of the Translational Perspective paragraph.
- 7) Papers must comply with the Statistics Policy: https://jp.msubmit.net/cgi-bin/main.plex?form_type=display_requirements#statistics.

In summary:

- If n {less than or equal to} 30, all data points must be plotted in the figure in a way that reveals their range and distribution. A bar graph with data points overlaid, a box and whisker plot or a violin plot (preferably with data points included) are acceptable formats.

- If $n > 30$, then the entire raw dataset must be made available either as supporting information, or hosted on a not-for-profit repository, e.g. FigShare, with access details provided in the manuscript.
- 'n' clearly defined (e.g. x cells from y slices in z animals) in the Methods. Authors should be mindful of pseudoreplication.
- All relevant 'n' values must be clearly stated in the main text, figures and tables.
- The most appropriate summary statistic (e.g. mean or median and standard deviation) must be used. Standard Error of the Mean (SEM) alone is not permitted.
- Exact p values must be stated. Authors must not use 'greater than' or 'less than'. Exact p values must be stated to three significant figures even when 'no statistical significance' is claimed.

8) Please include an Abstract Figure file, as well as the Figure Legend text within the main article file. The Abstract Figure is a piece of artwork designed to give readers an immediate understanding of the research and should summarise the main conclusions. If possible, the image should be easily 'readable' from left to right or top to bottom. It should show the physiological relevance of the manuscript so readers can assess the importance and content of its findings. Abstract Figures should not merely recapitulate other figures in the manuscript. Please try to keep the diagram as simple as possible and without superfluous information that may distract from the main conclusion(s). Abstract Figures must be provided by authors no later than the revised manuscript stage and should be uploaded as a separate file during online submission labelled as File Type 'Abstract Figure'. Please also ensure that you include the figure legend in the main article file. All Abstract Figures should be created using BioRender. Authors should use The Journal's premium BioRender account to export high-resolution images. Details on how to use and access the premium account are included as part of this email.

9) Please ensure that all figures and tables have a title and legend, and that they have been cited within the main article text.

EDITOR COMMENTS

The manuscript is methodologically rigorous. The novel comparative design addresses a gap in the literature and the findings presented are novel and physiologically significant, with implications for understanding brain energetics in health and disease. Fascinating work.

Precise P-values are not reported everywhere, particularly absent from the figures.

The discussion is well contextualised and highlights the translational implications for ageing, neurodegenerative disease and exercise physiology. While the study is limited by the absence of isotope tracers, this is acknowledged transparently and does not detract from the overall impact.

I have some comments to consider:

- Statistical presentation - please use precise p-values, including text, figures and tables. Please refer to the authors' guidelines for detailed guidance on this.
- Sample size - the study includes 13 participants, but there is no mention of how this number was determined. Was a sample size calculation performed?
- Translational relevance - the study population consists of young, healthy individuals, and the authors do discuss implications for older and clinical populations. To strengthen this further, I encourage a deeper exploration of how these findings might translate to impaired or diseased brains, where glucose metabolism is already compromised. Would the same lactate-glucose dynamics hold in populations with altered blood-brain barrier function or mitochondrial dysfunction? Additionally, could the authors discuss what therapeutic strategies might emerge from this work, such as lactate supplementation, exercise prescriptions or metabolic interventions?

One expert has reviewed the manuscript and also provided some feedback that is worth considering. I hope all the feedback provided will help you advance your work further and I thank you for considering The Journal of Physiology for your work.

Senior Editor:

Thank you for the submission of your research article to the Journal of Physiology for consideration. The article has been reviewed by experts in the field and found to require a major revision before any further decision can be made. Further consideration will require adequate revision to address all of the concerns raised. These concerns may require additional data and/or experiments. Please address all comments from the external referee and reviewing editor, as well as address

the list of requirements for publication in the journal, including the statistical requirements, as outlined in this letter, the Instructions to Authors published online, and the Journal's policy on rigour and reproducibility found at the link provided below.

<https://physoc.onlinelibrary.wiley.com/pb-assets/hub-assets/physoc/documents/TJP-Rigour-and-Reproducibility-Requirements-1724673661727.pdf>

For numerical, continuous (scalable) data, justify use of parametric or non-parametric tests. State the method(s) used to determine adequate sample size. Clearly state all relevant 'n' values in the main text, figures and/or tables.

Provide actual p values throughout, including figures. Do not use symbols.

State the statistical test(s) in the figure legends for comparisons shown in the figures.

The following formatting issues were found in the original manuscript and should be addressed. This list is not exclusive of the formatting requirements by the journal found above.

1. Begin the Methods with a section titled "Ethical Approval".

Include a statement that the protocol was approved, or declared exempt, by an appropriate governing body; please state the name of the approving authority. State the ethics approval number (or upload a letter of ethics approval with the manuscript). Include a statement that studies conformed to the standards set by the latest version of the Declaration of Helsinki (or the version that was in place at the time of the experiments). For clinical trials, additionally state that the studies conformed to clause 35 of the Declaration. State the range of demographic data as well as the mean (SD) (please declare the variance is SD). State inclusion and exclusion criteria.

2. Statistical Analysis

For numerical, continuous (scalable) data, justify the use of parametric or non-parametric tests. State the method(s) used to determine adequate sample size. Clearly state all relevant 'n' values in the main text, figures, and/or tables.

Provide actual p-values throughout, including figures. Do not use symbols.

State the statistical test(s) in the figure legends for comparisons shown in the figures.

REFEREE COMMENTS

Referee #2:

Under normal conditions, the resting brain is fueled by glucose with a small release of lactate. Thus, in the resting brain there is a net efflux of lactate, as shown by a negative arterio-venous difference in lactate (i.e., higher lactate concentration is measured in jugular vein compared to jugular artery). However, when circulating lactate is elevated, lactate oxidation increases. They tested if mechanism of glucose-sparing effect differs when lactate is elevated via passive infusion versus elevated via exercise.

In this paper entitled "Preferential lactate metabolism in the human brain during exogenous and endogenous hyperlactaemia" Koep et al did two experiments: with intravenous sodium-lactate infusion (exogenous hyperlactataemia) and with cycling exercise (endogenous hyperlactataemia). They observed that exogenous infusion protocol resulted in a higher CBF compared to exercise, despite some systemic alkalosis. During both protocols CMRO₂ remained unchanged across increases in lactate concentrations, while CMR_{glc} decreased and CMR_{lac} increased in a dose dependent manner. They conclude that elevated circulating lactate leads to preferential lactate oxidation and reduced glucose utilization, irrespective of whether lactate is delivered exogenously or produced endogenously. If increased brain carbohydrate uptake (i.e., glucose and lactate) is in excess to oxygen, it is termed aerobic glycolysis. But the destination of this lactate is unknown. They need to provide some comparison of theories of where lactate goes. The fate of CSF lactate needs to be considered. Is all the lactate and glucose accounted for fully? Please consider several hypotheses for the destination of these lost carbons from lactate or glucose:

- increased flux of glucose through the pentose phosphate pathway
- shunting of glucose through glycogen
- dispersal and loss of lactate through perivascular

Please discuss the estimated rates of these processes to see if any of these pathways (e.g., pentose phosphate pathway is known to be very slow as these are biosynthetic pathways) can be excluded in the short period post exercise.

Since the perivascular space is inextricably linked to the CSF space, its inclusion as a hypothesis seems contradictory because they measured lactate in CSF along with blood. To include the perivascular space as part of their hypotheses, they must include some additional literature support of this pathway.

The most plausible pathway may be the glycogen shunt. Please cite old and recent literature on glycogen shunt theory and discuss how much glycogen is likely to be mobilized with the unaccounted-for lactate.

END OF COMMENTS

REVISION CHECKLIST: REQUIRED ITEMS

1. COMMENT: Author photo and profile. First or joint first authors are asked to provide a short biography (no more than 100 words for one author or 150 words in total for joint first authors) and a portrait photograph. These should be uploaded and clearly labelled together in a Word document with the revised version of the manuscript. See Information for Authors for further details.

RESPONSE: This has now been included in Lines 44.

2. COMMENT: The reference list must be in alphabetical order, rather than numbered, to comply with our Journal format.

RESPONSE 2: The reference list and all in text citations have been amended according to journal formatting.

3. COMMENT: Please upload separate high-quality figure files via the submission form.

RESPONSE: Individual and high-quality figures have now been uploaded on the submission portal

4. COMMENT: Please ensure that any tables are editable and in Word format, and wherever possible, embedded in the article file itself.

RESPONSE: Data tables are included in the manuscript at the end of the file as an appendix. As highlighted in the below comment this was moved from a supplementary file to the main manuscript and is now cited in text in lines 305.

5. COMMENT: Your paper contains Supporting Information of a type that we no longer publish, including supplementary tables and figures. Any information essential to an understanding of the paper must be included as part of the main manuscript and figures. The only Supporting Information that we publish are video and audio, 3D structures, program codes and large data files. Your revised paper will be returned to you if it does not adhere to our Supporting Information Guidelines.

RESPONSE: We have revised the manuscript to now include the previously submitted supplementary Table and Supplementary figures in the main manuscript as an appendix. Table 1A is cited in the results (lines 304-306), and Figure A1 is cited in the discussion in lines 505. These are both included within the article file following the manuscript.

6. COMMENT: We invite you to include a Translational Perspective paragraph in your manuscript. This should be included in the main body of the manuscript after the Acknowledgements. It should describe the wider translational implications of the work, in plain English, for a broad scientific audience. Please use the following guidelines to prepare a

Translational Perspective of your paper: https://jp.msubmit.net/cgi-bin/main.plex?form_type=display_requirements#authortranspersp. The Translational Perspective

should not exceed 250 words in total and should be presented as a single paragraph. Abbreviations and technical terms must be defined as briefly and simply as possible the first time they are used, unless they are generally/easily understood, e.g. ECG, HIV/AIDS, K+ channel. Use language that can be understood by scientists or clinicians with a general knowledge of the topic addressed. Ensure the paragraph includes the hypothesis tested in the paper and accurately reflects the findings of the paper and the implications

for future research. Please state the word count of the Translational Perspective paragraph.

RESPONSE: A single paragraph 250-word translational perspective has now been added in line with journal recommendations emphasizing the translational potential of this work. This can be seen in lines 560-579 and reads as follows: *“This study tested whether the human brain preferentially uses lactate over glucose as fuel when circulating lactate levels are elevated. Using two different methods - vigorous exercise and passive lactate infusion, we demonstrated that increased lactate availability causes the brain to reduce its glucose consumption by up to 60%, indicating that lactate becomes the preferred energy source. This glucose-sparing effect occurred regardless of whether lactate was elevated through exercise or passive infusion, suggesting a robust and metabolic preference of the brain for lactate under conditions of elevated supply. Despite increased fuel availability, brain oxygen uptake remained unchanged, suggesting that the preserved glucose is directed toward non-oxidative metabolic pathways that support critical brain functions including stress resistance, cellular repair, and energy storage. These findings offer important insight for neurological conditions where glucose metabolism is impaired, such as Alzheimer's disease, traumatic brain injury, and diabetes-related cognitive decline (Baker et al., 2011; Chen & Zhong, 2013; Costantini et al., 2008). Since lactate can effectively substitute glucose for brain energetic needs, therapeutic strategies that safely elevate lactate levels, whether through controlled exercise, or pharmacological approaches, could maintain brain energy supply when glucose metabolism is compromised. While disease-specific metabolic responses may differ, the brain's apparent metabolic flexibility to use alternative fuels could partly explain why exercise provides cognitive benefits and neuroprotection. Future research should investigate optimal lactate dosing strategies, explore lactate-based therapies for neurodegenerative diseases, and determine whether lactate supplementation can preserve cognitive function in at-risk populations, potentially opening new avenues for treating brain metabolic disorders.”*

7. RESPONSE: Papers must comply with the Statistics Policy: https://jp.msubmit.net/cgi-bin/main.plex?form_type=display_requirements#statistics.

In summary:

- If $n \leq 30$

30, all data points must be plotted in the figure in a way that reveals their range and distribution. A bar graph with data points overlaid, a box and whisker plot or a violin plot (preferably with data points included) are acceptable formats.

- If $n > 30$, then the entire raw dataset must be made available either as supporting information, or hosted on a not-for-profit repository, e.g. FigShare, with access details provided in the manuscript.

- 'n' clearly defined (e.g. x cells from y slices in z animals) in the Methods. Authors should be mindful of pseudoreplication.

- All relevant 'n' values must be clearly stated in the main text, figures and tables.

- The most appropriate summary statistic (e.g. mean or median and standard deviation) must be used. Standard Error of the Mean (SEM) alone is not permitted.

- Exact p values must be stated. Authors must not use 'greater than' or 'less than'. Exact p values must be stated to three significant figures even when 'no statistical significance' is claimed.

RESPONSE: Amendments have been made to ensure statistics are reported in line with journal recommendations, including exact p values throughout reported to 3 significant figures.

8. COMMENT: Please include an Abstract Figure file, as well as the Figure Legend text within the main article file. The Abstract Figure is a piece of artwork designed to give

readers an immediate understanding of the research and should summarise the main conclusions. If possible, the image should be easily 'readable' from left to right or top to bottom. It should show the physiological relevance of the manuscript so readers can assess the importance and content of its findings. Abstract Figures should not merely recapitulate other figures in the manuscript. Please try to keep the diagram as simple as possible and without superfluous information that may distract from the main conclusion(s). Abstract Figures must be provided by authors no later than the revised manuscript stage and should be uploaded as a separate file during online submission labelled as File Type 'Abstract Figure'. Please also ensure that you include the figure legend in the main article file. All Abstract Figures should be created using BioRender. Authors should use The Journal's premium BioRender account to export high-resolution images. Details on how to use and access the premium account are included as part of this email.

RESPONSE: The abstract figure has been created and uploaded as a separate file, and the figure abstract legend can be seen in the main article file in lines 70-80.

9. COMMENT: Please ensure that all figures and tables have a title and legend, and that they have been cited within the main article text.

RESPONSE: All figures have been amended to ensure they have a clear title, and a legend which details necessary information including statistical analyses.

EDITOR COMMENTS

1. COMMENT: The manuscript is methodologically rigorous. The novel comparative design addresses a gap in the literature and the findings presented are novel and physiologically significant, with implications for understanding brain energetics in health and disease. Fascinating work.

RESPONSE: Thank you for the supportive comments, and the opportunity to amend the manuscript in line with your feedback.

2. COMMENT: Precise P-values are not reported everywhere, particularly absent from the figures.

RESPONSE: This has been addressed to ensure all tables, figures and in text display exact P values, and that p values are reported to 3 significant figures.

3. COMMENT: The discussion is well contextualised and highlights the translational implications for ageing, neurodegenerative disease and exercise physiology. While the study is limited by the absence of isotope tracers, this is acknowledged transparently and does not detract from the overall impact.

RESPONSE: Thank you for your feedback and the opportunity to revise our manuscript in line with recommendations. We appreciate the acknowledgment that the present data provide a novel and important advancement despite the methodological limitation.

I have some comments to consider:

4. COMMENT: Statistical presentation - please use precise p-values, including text, figures and tables. Please refer to the authors' guidelines for detailed guidance on this.

5. COMMENT: Sample size - the study includes 13 participants, but there is no mention of how this number was determined. Was a sample size calculation performed?

RESPONSE: Given the constraints of an invasive study, this was a sample size of convenience with no *a priori* power calculation. Effect sizes, however, confirm large effect sizes for the primary outcome measures of changes in CMRglc with increasing arterial lactate concentrations. This has been added to lines 286-288 in the manuscript and reads as follows: “Sample size (n=13) was determined by practical constraints given the invasive nature of the study protocol. Large effect sizes for primary outcomes of interest supported adequate power to detect significant effects (CMRglc, $np^2=0.141$) (Cohen 1992).”

6. COMMENT: Translational relevance - the study population consists of young, healthy individuals, and the authors do discuss implications for older and clinical populations. To strengthen this further, I encourage a deeper exploration of how these findings might translate to impaired or diseased brains, where glucose metabolism is already compromised. Would the same lactate-glucose dynamics hold in populations with altered blood-brain barrier function or mitochondrial dysfunction? Additionally, could the authors discuss what therapeutic strategies might emerge from this work, such as lactate supplementation, exercise prescriptions or metabolic interventions?

RESPONSE: Thank you for raising these translational considerations which strengthen the potential impact of the work. We have added discussion acknowledging that neurodegenerative diseases commonly exhibit mitochondrial dysfunction and altered glucose metabolism, and that changes in monocarboxylate transporter expression and BBB permeability could affect lactate-glucose dynamics. We also discuss potential therapeutic strategies while noting the need for future research in clinical populations. These discussions are integrated into the implication section, and as follows in lines 543: “An important consideration for future investigations into the therapeutic potential of lactate supplementation is whether its efficacy is maintained in neurological disease states characterized by BBB disruption, altered MCT expression, and mitochondrial dysfunction (Bergersen, 2015; He et al., 2025). Elucidating these factors may inform targeted strategies, such as modulation of MCT expression or mitochondrial capacity to optimize lactate utilization in pathological conditions” Further, in line with recommendations we have added a translational perspectives paragraph in lines 574 which largely focusses on the application of this work to disease states.

7. COMMENT: One expert has reviewed the manuscript and also provided some feedback that is worth considering. I hope all the feedback provided will help you advance your work further and I thank you for considering The Journal of Physiology for your work.

RESPONSE: Thank you for your comments and review of the present work, which we believe to have strengthened the overall manuscript, and in particular the translational relevance and applications of this work. We hope you find the responses and edits to the manuscript satisfactory.

Senior Editor:

1. COMMENT: Thank you for the submission of your research article to the Journal of Physiology for consideration. The article has been reviewed by experts in the field and found to require a major revision before any further decision can be made. Further consideration will require adequate revision to address all of the concerns raised. These concerns may require additional data and/or experiments. Please address all comments from the external referee and reviewing editor, as well as address the list of requirements for publication in the journal, including the statistical requirements, as outlined in this

letter, the Instructions to Authors published online, and the Journal's policy on rigour and reproducibility found at the link provided below. <https://physoc.onlinelibrary.wiley.com/pb-assets/hub-assets/physoc/documents/TJP-Rigour-and-Reproducibility-Requirements-1724673661727.pdf>

RESPONSE: Thank you for your review of our work, and for the opportunity to revise the present manuscript in line with reviewer comments and journal guidelines. We hope you find our responses to be satisfactory and to address any concerns raised.

2. COMMENT: For numerical, continuous (scalable) data, justify use of parametric or non-parametric tests. State the method(s) used to determine adequate sample size. Clearly state all relevant 'n' values in the main text, figures and/or tables.

These missing methods details have now been added into the respective sections. In the statistical analysis section, the confirmation of parametric test assumptions has been detailed as follows in lines 289-291: "Data were normally distributed as assessed by visual inspection of Q-Q plots and homoscedasticity of the studentized residuals plotted against the predicted values. Linearity of relationships was established by visual inspection of scatterplots." For justification of the sample size, given the constraints of an invasive study, this was a convenience sample with no a priori power calculation. Effect sizes however, confirm large effect sizes for the primary outcome measures of changes in CMRglc with increasing arterial lactate concentrations. This has been added to lines 286-289 in the manuscript and reads as follows: "Sample size (n=13) was determined by practical constraints given the invasive nature of the study protocol. Large effect sizes for primary outcomes of interest supported adequate power to detect significant effects (CMRglc, $np^2=0.141$) (Cohen 1992)." Regarding the requirement to clearly state all relevant 'n' values, we have reviewed and ensured this is explicitly stated in all analyses throughout the main text, figures, and tables where statistical comparisons are presented. Additions to the figures read as follows: "For the passive infusion, n=13 for all conditions (0, 4, and 8 mmol/L), and for the exercise protocol n=13 at 0mmol/L, n=10 at 4 mmol/L, and n=12 at 8 mmol/L"

3. COMMENT: Provide actual p values throughout, including figures. Do not use symbols.

RESPONSE: This has been addressed to ensure all tables, figures and in text display exact P values, and that p values are reported to 3 significant figures throughout.

4. COMMENT: State the statistical test(s) in the figure legends for comparisons shown in the figures.

RESPONSE: This has been addressed and all figures contain details of statistical analyses displayed.

5. COMMENT: The following formatting issues were found in the original manuscript and should be addressed. This list is not exclusive of the formatting requirements by the journal found above.

- Begin the Methods with a section titled "Ethical Approval".

Include a statement that the protocol was approved, or declared exempt, by an appropriate governing body; please state the name of the approving authority. State the ethics approval number (or upload a letter of ethics approval with the manuscript).

Include a statement that studies conformed to the standards set by the latest version of the Declaration of Helsinki (or the version that was in place at the time of the experiments). For clinical trials, additionally state that the studies conformed to clause

35 of the Declaration. State the range of demographic data as well as the mean (SD) (please declare the variance is SD). State inclusion and exclusion criteria.

RESPONSE: Thank you for drawing our attention to this requirement. This section has now been added in Lines 121 to 129 as follows: "Ethical Approval This study was approved by the clinical research ethics board of the University of British Columbia (#H23-02303). All experimental protocols and procedures conformed to the standards set by the Canadian government Tri-Council policy statement for integrity in research, as well as the declaration of Helsinki, including registration in a database (approval #77764). A detailed verbal and written explanation of the procedures and measurements was provided to participants prior to providing written, informed consent. Experiments were conducted with medical support and by suitably qualified personnel. Healthy normotensive volunteers who did not require daily medication (excluding contraceptive medications) were recruited for the study. Exclusion criteria included current or former smokers, a known history of cardiometabolic or respiratory diseases, and the use of medications. Thirteen aerobically fit [maximal oxygen uptake, ($\dot{V}O_{2max}$): 46.9 ± 6.2 mL/min/kg] participants (age 28.2 ± 3.5 yrs, 6 females) were recruited."

- Statistical Analysis

Provide actual p-values throughout, including figures. Do not use symbols.

RESPONSE: This has been addressed to ensure all tables, figures and in text display exact P values, and that p values are reported to 3 significant figures.

- State the statistical test(s) in the figure legends for comparisons shown in the figures.

RESPONSE: Figures all now contain the relevant detail for interpretation of statistical comparisons shown.

REFEREE COMMENTS

Referee #2:

COMMENT: Under normal conditions, the resting brain is fueled by glucose with a small release of lactate. Thus, in the resting brain there is a net efflux of lactate, as shown by a negative arterio-venous difference in lactate (i.e., higher lactate concentration is measured in jugular vein compared to jugular artery). However, when circulating lactate is elevated, lactate oxidation increases. They tested if mechanism of glucose-sparing effect differs when lactate is elevated via passive infusion versus elevated via exercise.

In this paper entitled "Preferential lactate metabolism in the human brain during exogenous and endogenous hyperlactaemia" Koep et al did two experiments: with intravenous sodium-lactate infusion (exogenous hyperlactataemia) and with cycling exercise (endogenous hyperlactataemia). They observed that exogenous infusion protocol resulted in a higher CBF compared to exercise, despite some systemic alkalosis. During both protocols CMRO₂ remained unchanged across increases in lactate concentrations, while CMR_{glc} decreased and CMR_{lac} increased in a dose dependent manner. They conclude that elevated circulating lactate leads to preferential lactate oxidation and reduced glucose utilization, irrespective of whether lactate is delivered exogenously or produced endogenously. If increased brain carbohydrate uptake (i.e., glucose and lactate) is in excess to oxygen, it is termed aerobic glycolysis. But the destination of this lactate is unknown. They need to provide some comparison of theories of where lactate goes. The fate of CSF lactate needs to be considered. Is all the

lactate and glucose accounted for fully? Please consider several hypotheses for the destination of these

lost carbons from lactate or glucose:

- increased flux of glucose through the pentose phosphate pathway
- shunting of glucose through glycogen
- dispersal and loss of lactate through perivascular

Please discuss the estimated rates of these processes to see if any of these pathways (e.g., pentose phosphate pathway is known to be very slow as these are biosynthetic pathways) can be excluded in the short period post exercise.

Since the perivascular space is inextricably linked to the CSF space, its inclusion as a hypothesis seems contradictory because they measured lactate in CSF along with blood. To include the perivascular space as part of their hypotheses, they must include some additional literature support of this pathway.

The most plausible pathway may be the glycogen shunt. Please cite old and recent literature on glycogen shunt theory and discuss how much glycogen is likely to be mobilized with the unaccounted-for lactate.

RESPONSE: Thank you for your supportive comments and raising this important point regarding the fate of excess carbohydrate uptake during hyperlactatemia, which requires an explanation as to where the excess carbons are directed. As you can appreciate, especially in a human model, it is difficult (if not impossible) to directly isolate and estimate rates/fluxes of each pathway. Under the present experimental conditions, a proportion of glucose extracted by the brain is not oxidized or released as lactate, indicating a loss of carbons through non measurable outputs. As noted, while multiple pathways, inclusive of 1) the pentose phosphate pathway, 2) glycogen shunt, and 3) perivascular clearance may contribute, no single mechanism appears to fully account for the observed glucose sparring.

Pentose phosphate pathway (PPP): The PPP represents a glucose-oxidizing pathway that branches from glycolysis at glucose-6-phosphate, serving critical biosynthetic and antioxidant functions (Baquer et al., 1988; Hothersall et al., 1982). Although we cannot provide an accurate estimate of PPP flux in the whole brain during our experimental conditions, exercise is known to upregulate PPP activity in an acute setting, providing a plausible explanation for the consumed but non-oxidized glucose (Dienel et al 2019). Importantly, since PPP utilizes glycolytic intermediates (glucose-6-phosphate), increased lactate oxidation during exercise or passive infusion may allow glucose to be preferentially directed through this energetically-inefficient but metabolically important pathway. However, given that PPP represents a relatively slow biosynthetic pathway, it is unlikely to account for the substantial glucose consumption observed in the acute post-exercise period.

Glycogen synthesis: Brain glycogen turnover represents another potential destination for the excess glucose uptake. The glycogen shunt hypothesis suggests that glucose may be temporarily stored as glycogen before being mobilized for oxidative metabolism (Dienel, 2019; Matsui, 2021; Matsui et al., 2012). This pathway is particularly relevant given that exhaustive exercise in habituated rats decreases whole brain glycogen concentration by approximately 40% (Matsui et al., 2020), suggesting that circulating glucose may be continuously replenishing brain glycogen stores during and following exercise, thereby contributing to the observed decrease in OCI. The glycogen shunt represents the most temporally plausible mechanism to

account for the substantial glucose consumption observed in our study, as glycogen synthesis can occur rapidly and accommodate significant glucose flux.

Perivascular clearance: A third hypothesis involves the dispersal and clearance of lactate and glucose through perivascular pathways. During strenuous exercise, increased consumption of oxygen and glucose, along with metabolic water produced by oxidative metabolism and water release from glycogen breakdown, may enhance fluid fluxes from intracellular sources to interstitial fluid and subsequently to lymphatic drainage systems (Dienel & Lauritzen, 2025). This enhanced clearance could theoretically account for some of the unmetabolized glucose. However, since we measured lactate concentrations in both blood and CSF and the perivascular space is inextricably linked to the CSF space, this pathway seems less likely to fully explain our observations without additional evidence of differential clearance rates between these compartments.

While trying to avoid too much speculation, we have added relevant details to the current discussion section to emphasise these pathways and estimated contributions of glucose in which they account for. These changes and additions to the manuscript can be seen in lines 441 and read as follows: “*These glucose-derived glycolysis intermediates may subserve alternative fates beyond that of energy production via conversion to pyruvate or lactate (Benveniste et al., 2018; Dienel, 2019). Although the present experimental model cannot directly address the fate of this excess glucose, several potential pathways warrant discussion. These include diversion into the pentose phosphate pathway (PPP) (managing oxidative stress) (Baquer et al., 1988; Hothersall et al., 1982), serving as a precursor for several structural compounds (Jones et al., 1975; Ramsey et al., 1971), storage via the glycogen shunt (Dienel, 2019; Matsui et al., 2012; Matsui, 2021) or dispersal through perivascular-lymphatic clearance (Dienel & Lauritzen, 2025). We propose the most likely explanation is that lactate oxidation spares glucose for PPP and storage as glycogen (Dienel et al., 2002), while simultaneously some lactate is released from the brain via perivascular drainage (Ball et al., 2010). This glucose sparing effect may be reflective of lactates reflects more constrained metabolic fate.*”

Given this is speculative and outside the scope of the present experimental model and research question, we have kept discussions concise, whilst acknowledging these hypotheses. We hope you find these additions satisfactory and to have addressed this point clearly.

END OF COMMENTS

Dear Dr Koep,

Re: JP-RP-2025-289216R1 "**Preferential lactate metabolism in the human brain during exogenous and endogenous hyperlactatemia**" by Jodie Lauren Koep, Jennifer Duffy, Jay M.J.R. Carr, L. Madden Brewster, Jordan Douglas Bird, Justin A Monteleone, Tenasia Monaghan, Hashim Islam, Andrew Robert Steele, Connor A Howe, David B MacLeod, Philip N Ainslie, and Travis D Gibbons

Thank you for submitting your manuscript to The Journal of Physiology. It has been assessed by a Reviewing Editor and by 1 expert referee and we are pleased to tell you that it is acceptable for publication following satisfactory revision.

REVISION CHECKLIST:

Please upload two versions of your manuscript text: one with all relevant changes highlighted and one clean version with no changes tracked. The manuscript file should include all tables and figure legends, but each figure/graph should be uploaded as separate, high-resolution files. The journal is now integrated with Wiley's Image Checking service. For further details,

see: <https://www.wiley.com/en-us/network/publishing/research-publishing/trending-stories/upholding-image-integrity-wileys-image-screening-service>

We look forward to receiving your revised submission.

Yours sincerely,

Harold Schultz
Senior Editor
The Journal of Physiology

REQUIRED ITEMS

1) - Papers must comply with the Statistics Policy: https://jp.msubmit.net/cgi-bin/main.plex?form_type=display_requirements#statistics.

- Exact p values must be stated. Authors must not use 'greater than' or 'less than'. Exact p values must be stated to three significant figures even when 'no statistical significance' is claimed.

EDITOR COMMENTS

Reviewing Editor:

Thank you for providing your response and for addressing the previous comments.

We are nearly there - just a few minor points to consider:

- It appears that the precise p-values are missing in Figures 2, 3 and A5. Currently, only significance stars are shown.
- Additionally, the reviewer has a comment regarding a reference included in the manuscript.

Senior Editor:

Thank you for the submission of your revised research article to the Journal of Physiology for consideration. The article has been reviewed as acceptable for publication pending adequate revision to address a remaining concern from the referee and statistical requirements in the figures.

The precise p-values are missing in Figures 2, 3 and A5. Currently, only significance stars are shown.

REFeree COMMENTS

Referee #2:

Thanks for revising the text to reflect the mechanisms of unmetabolized lactate. However, it would be great if it can be explicitly stated that PPP represents a relatively slow biosynthetic pathway compared to the other mechanisms mentioned. Furthermore, the original glycogen shunt in brain were formalized by Shuman et al (2001) in PNAS and NMR Biomed.

END OF COMMENTS

REQUIRED ITEMS

1. Papers must comply with the Statistics Policy: https://jp.msubmit.net/cgi-bin/main.plex?form_type=display_requirements#statistics.

- Exact p values must be stated. Authors must not use 'greater than' or 'less than'. Exact p values must be stated to three significant figures even when 'no statistical significance' is claimed.

Apologies for missing these P values on the last amendment, the exact p values reported to 3 significant figures have now been added to figures 2, 3 and A5.

EDITOR COMMENTS

Reviewing Editor:

1. Thank you for providing your response and for addressing the previous comments.

We are nearly there - just a few minor points to consider:

- It appears that the precise p-values are missing in Figures 2, 3 and A5.

Currently, only significance stars are shown.

Thank you for noting these figures whereby precise p values are missing, this has now been amended.

2. Additionally, the reviewer has a comment regarding a reference included in the manuscript.

This has now been addressed to add the recommended information and reference.

Senior Editor:

1. Thank you for the submission of your revised research article to the Journal of Physiology for consideration. The article has been reviewed as acceptable for publication pending adequate revision to address a remaining concern from the referee and statistical requirements in the figures.

- The precise p-values are missing in Figures 2, 3 and A5. Currently, only significance stars are shown.

Thank you for noting these figures whereby precise p values are missing, this has now been amended.

REFEREE COMMENTS

Referee #2:

1. Thanks for revising the text to reflect the mechanisms of unmetabolized lactate. However, it would be great if it can be explicitly stated that PPP represents a relatively slow biosynthetic pathway compared to the other mechanisms mentioned. Furthermore, the original glycogen shunt in brain were formalized by Shuman et al (2001) in PNAS and NMR Biomed.

We have now revised the text to explicitly state that the pentose phosphate pathway represents a relatively slow biosynthetic route compared with other mechanisms. In addition, we now cite Shulman (2001, *PNAS*) to acknowledge the initial glycogen shunt concept in the brain. These revisions can be seen in lines 439 and 442.

END OF COMMENTS

Dear Dr Koep,

Re: JP-RP-2025-289216R2 "**Preferential lactate metabolism in the human brain during exogenous and endogenous hyperlactatemia**" by Jodie Lauren Koep, Jennifer Duffy, Jay M.J.R. Carr, L. Madden Brewster, Jordan Douglas Bird, Justin A Monteleone, Tenasia Monaghan, Hashim Islam, Andrew Robert Steele, Connor A Howe, David B MacLeod, Philip N Ainslie, and Travis D Gibbons

We are pleased to tell you that your paper has been accepted for publication in The Journal of Physiology.

Yours sincerely,

Harold Schultz
Senior Editor
The Journal of Physiology

If you would like to receive our 'Research Roundup', a monthly newsletter highlighting the cutting-edge research published in The Physiological Society's family of journals (The Journal of Physiology, Experimental Physiology, Physiological Reports, The Journal of Nutritional Physiology and The Journal of Precision Medicine: Health and Disease), please click this link, fill in your name and email address and select 'Research Roundup':
<https://www.physoc.org/journals-and-media/membernews>

- You can help your research get the attention it deserves! Check out Wiley's free Promotion Guide for best-practice recommendations for promoting your work at: www.wileyauthors.com/eoo/guide. You can learn more about Wiley Editing Services which offers professional video, design, and writing services to create shareable video abstracts, infographics, conference posters, lay summaries, and research news stories for your research at: www.wileyauthors.com/eoo/promotion.

EDITOR COMMENTS

Senior Editor:

The editors thank the authors for the final adjustments to the manuscript. The article is now accepted for publication.

Congratulations on an interesting and insightful study. Please consider the Journal of Physiology for your future studies.

REFeree COMMENTS